# Self-Consistent Flow: Unifying Velocity and Endpoint Prediction for Rectified Flow Models

**Xu Han**                                                                     *xu.han@tufts.edu*
*Department of Computer Science*
*Tufts University*

**Jiajing Hu**                                                              *jiajing.hu@tufts.edu*
*Department of Computer Science*
*Tufts University*

**Li-Ping Liu**                                                            *liping.liu@tufts.edu*
*Department of Computer Science*
*Tufts University*

**Reviewed on OpenReview:** *https://openreview.net/forum?id=z9FMXOJgyl*

## Abstract

In rectified-flow–based generative models, the neural network can be trained to predict two different targets, such as the instantaneous velocity or the data endpoint, to perform denoising. Although prior work shows that these parameterizations lead to different empirical behaviors, the mechanisms underlying their respective advantages remain to be underexplored, and how to combine them effectively is still unclear. In this work, we analyze how learning errors from different parameterizations affect the generation performance. We show that predicting the data endpoint has a clear training signal that stabilizes training, whereas predicting the velocity maintains stable sampling dynamics near the data manifold. Motivated by these insights, we propose Self-Consistent Flow (SC-Flow), a new method that unifies the benefits of both parameterizations. By employing a lightweight consistency loss, SC-Flow jointly trains a single network to predict *both* the local velocity and the data endpoint, and the consistency between the two predictions improves the model's performance. The method requires no major architectural changes and adds minimal computational overhead. Extensive experiments on image generation tasks demonstrate that SC-Flow substantially stabilizes optimization and improves the straightness of generation paths, leading to significant gains in generation quality over standard rectified-flow baselines.

## 1 Introduction

Generative models based on ordinary differential equations (ODEs) (Lipman et al., 2022; Liu et al., 2022), have emerged as a powerful alternative to traditional diffusion models, enabling high-quality synthesis with significantly fewer sampling steps. The neural network learns the vector field of an ODE that transforms a simple prior distribution into the data distribution.

In a flow matching model (Lipman et al., 2022; Liu et al., 2022), the generative model needs to predict the vector field to learn the ODE that defines the model distribution. Besides predicting the velocity field at a given noisy sample, the underlying neural network can also be parameterized to predict the clean sample or the noise (Li & He, 2025). Different parameterizations also appear in different diffusion-based generation (Song et al., 2020b), where the neural network can learn the score function in the reverse SDE

(Song & Ermon, 2019), predict the noise (Ho et al., 2020; Rombach et al., 2022), or predict the clean sample (Song et al., 2020a). While these different predicting targets are mathematically equivalent, they do yield distinct empirical performances. Previous studies that develop specific methods often consider that one parameterization is superior over another one depending on the empirical performance. Recent research (Li & He, 2025) compares different flow parameterization in a systematic study and favors X-Flow based on an argument of data manifold. Given that different parameterizations are equivalent in the optimal case, we believe that the learning error is the root reason for their performance difference. However, there still lacks a systematic analysis of learning errors in these different parameterizations.

In this work, we relate prediction targets to learning errors and develop a dual-target predictive model to reduce the error. We introduce Self-Consistent Flow (**SC-Flow**) to combine the strengths of the two learning targets. SC-Flow is a novel framework that trains a single model to simultaneously predict two complementary targets: the instantaneous velocity ($v$) and the likely destination of the ODE path ($x$). Our key insight is that by receiving training signals in both forms, the neural network gains extra power in learning the vector field, bridging the optimization benefits of data prediction with the inference stability of velocity prediction. SC-Flow uses a single network backbone to compute the two fitting targets, with a binary mode flag to indicate the type of the target. The design adds negligible overhead to a baseline rectified flow model and incurs the same level of inference costs in the generation procedure.

Experiments on ImageNet show that SC-Flow substantially outperforms baseline methods, demonstrating the advantage of combining flow targets in a unified model. We also conduct an extensive empirical study to verify our theoretical analysis and demonstrate that the performance gain is due to our new design.

Our contributions can be summarized as follows:

- **Dual-Target Framework:** We propose SC-Flow, a novel training framework for rectified flows that supervises a single model on two complementary targets: the instantaneous velocity and the likely endpoint. The consistency loss enforces the analytical relationship between the two predictions. This improves optimization and ensures the learned vector fields are coherent.

- **Analysis of learning errors:** We rigorously analyze the variance and error dynamics of flow estimators. We show that direct endpoint prediction is less affected by sampling variance, whereas the velocity target is more stable near the data manifold.

- **Empirical verification:** SC-Flow substantially outperforms the standard rectified flow baseline on ImageNet $256 \times 256$, achieving significant FID improvements with the same sampling cost and negligible additional training overhead.

## 2 Related Works

Modern denoising-based generative models are formulated through three closely related perspectives: score-based modeling (Song & Ermon, 2019), reverse diffusion processes (Song et al., 2020b; Ho et al., 2020), and flow matching (Liu et al., 2022; Lipman et al., 2022). Across these formulations, the neural networks are often parameterized differently to perform the denoising step. In score-based methods (Song & Ermon, 2019), the neural network is trained to learn the score function at different noise levels. The learning target is also equivalent to learning the score function in the reverse SDE (Song et al., 2020b). In DDPM (Ho et al., 2020), the network is trained to predict the noise ($\epsilon$-prediction). In DDIM (Song et al., 2020a), the network directly predicts the clean sample conditioned on the noisy sample.

In the framework of Flow Matching (Lipman et al., 2022) and Rectified Flows (Liu et al., 2022), the network learns a continuous-time ODE vector field to map a simple prior to the data distribution. With a noisy sample as the input, the output of the network should predict velocity ($v$-prediction) of the vector field. Compared against the diffusion-based formulation, the velocity is essentially the sum of the score function and the drift in the reverse SDE. With the assumption of rectified flow, the network can also predict the real data sample (X-Flow) or the noise ($\epsilon$-Flow), and then derive the velocity prediction. Therefore, X-Flow and $\epsilon$-Flow share a similar spirit respectively as DDPM and DDIM.

The different parameterizations are systematically investigated by Ma et al. (2024); Li & He (2025), which compared noise, score, and velocity predictions within the stochastic interpolant framework, demonstrating that generative performance heavily depends on the chosen target. In particular, Ma et al. (2024) observes a significant performance increase with weighted score and velocity (they are equivalent) in the latent diffusion model; while the study by Li & He (2025) favors $x$-flow on the original pixel space with an argument that the data is on a manifold and easy to predict. However, the argument is only supported by a small example. In this work, we deepen the understanding from a theoretical perspective and investigate how learning error affects the generation quality.

In diffusion-based generative models, Benny & Wolf (2022) have explored multiple parameterizations and dual outputs and let the neural network to both noise and the clean sample using a naive shared structure and a learned mixing strategy. However, this approach only relies on a shared network trunk and encourages the network to learn information from both targets. In our work, we develop a new algorithm from rectified flow, which is very different from diffusion-based models. Furthermore, we restrict the model to be consistent with the two flows.

## 3 Background

Rectified Flow (RF) learns a neural ODE to transform an initial noise distribution $p_0$ into the target data distribution $p_1$, with a time-dependent vector field $v_t(\mathbf{x}_t) : \mathbb{R}^d \rightarrow \mathbb{R}^d$ that guides the evolution of state $\mathbf{x}_t$:

$$d\mathbf{x}_t = v_t(\mathbf{x}_t)dt, \quad \mathbf{x}_0 \sim p_0, \quad t \in [0, 1]. \tag{1}$$

Starting from $\mathbf{x}_0 \sim p_0$ (typically a standard Gaussian distribution), RF integrates the ODE from $t = 0$ to $t = 1$, yielding a sample $\mathbf{x}_1 \sim p_1$. The vector field $v_t(\mathbf{x}_t)$ is learned by a neural network $\mathbf{v}_\theta(\mathbf{x}_t, t)$, where $\theta$ represents the trainable parameters.

In an RF model, the vector field is given by a *conditional* flow. RF starts with a coupling distribution $p(\mathbf{x}_0, \mathbf{x}_1)$ such that the marginal $p(\mathbf{x}_0)$ and $p(\mathbf{x}_1)$ are respectively the noise distribution and the data distribution. Then the conditional flow is defined by the straight path between $(\mathbf{x}_0, \mathbf{x}_1) \sim p(\mathbf{x}_0, \mathbf{x}_1)$. Let $\mathbf{x}_t = t\mathbf{x}_1 + (1 - t)\mathbf{x}_0$, the conditional velocity at $x_t$ is

$$u(\mathbf{x}_t | \mathbf{x}_0, \mathbf{x}_1) = \mathbf{x}_1 - \mathbf{x}_0 = \frac{\mathbf{x}_1 - \mathbf{x}_t}{1 - t}. \tag{2}$$

The model $\mathbf{v}_\theta(\mathbf{x}_t, t)$ is then trained with conditional flow loss by:

$$\mathcal{L}(\phi) = \mathbb{E}_{t \sim \mathcal{U}[0,1], \mathbf{x}_0, \mathbf{x}_1} \left[ \|(\mathbf{x}_1 - \mathbf{x}_0) - \mathbf{v}_\theta(\mathbf{x}_t, t)\|_2^2 \right]. \tag{3}$$

Latent space diffusion/flow models achieve both high generation quality and efficiency compared to the pixel space models. In the latent space model, the RGB image is encoded to a latent space sample $\mathbf{x}$ using an encoder $E$. The reconstructed image is decoded from the latent $\mathbf{x}$ by the decoder $D$. In this work, we focus on learning the data distribution $p(\mathbf{x}_1)$ and assume the encoder-decoder is fixed.

## 4 SC-Flow Model

**Motivation.** SC-Flow is motivated by re-examining the flow-matching objective. Let $p(\mathbf{x}_0, \mathbf{x}_1) = p_0(\mathbf{x}_0)p_1(\mathbf{x}_1)$ denote the independent coupling between the prior and the target data distribution. Here, $\mathbf{x}_1 \sim p_1$ is a clean sample drawn from the empirical data distribution, and $\mathbf{x}_0 \sim p_0 = \mathcal{N}(\mathbf{0}, \mathbf{I})$ is a random noise sample drawn from a standard Gaussian prior.

By the definition of the straight path $\mathbf{x}_t = (1 - t)\mathbf{x}_0 + t\mathbf{x}_1$, we can algebraically rewrite the instantaneous velocity as $\mathbf{x}_1 - \mathbf{x}_0 = \frac{\mathbf{x}_1 - \mathbf{x}_t}{1 - t}$. Substituting this directly into the flow-matching objective, for $t \in (0, 1)$, the

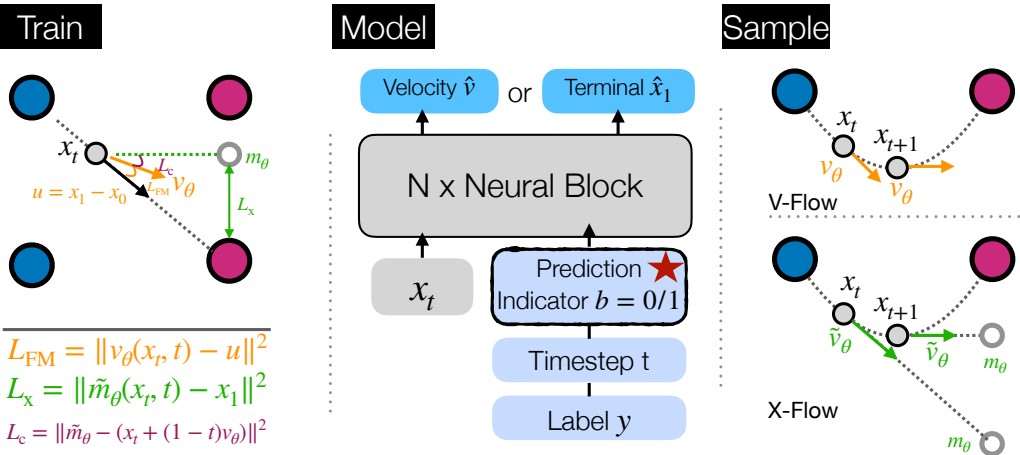

Figure 1: **SC-Flow overview: (A) Training.** Linear rectified path $x_t = (1-t)x_0 + tx_1$ with instantaneous velocity $u = x_1 - x_0$; Training minimizes $L_{FM} = \|\mathbf{v}_\theta - u\|^2$, $L_x = \|\tilde{\mathbf{m}}_\theta - x_1\|^2$, and the consistency term $L_c = \|\tilde{\mathbf{m}}_\theta - (x_t + (1-t)\mathbf{v}_\theta)\|^2$. **(B) Model.** Single network $\mathrm{nn}_\theta(x_t, t, y, b)$ with summed embeddings $E_t(t) + E_y(y) + E_b(b)$; mode $b{=}0$ outputs $\mathbf{v}_\theta$ (velocity) and $b{=}1$ outputs $\tilde{\mathbf{m}}_\theta$ (endpoint). **(C) Sampling.** Sampling integrates the probability-flow ODE with a per-step drift chosen from $\mathbf{v}_\theta$ and $(\tilde{\mathbf{m}}_\theta - x_t)/(1-t)$ (Equation (10)); any selection/blending policy may be used without retraining.

training loss becomes

$$\mathbb{E}_p\left[\|\mathbf{v}_\theta(\mathbf{x}_t, t) - (\mathbf{x}_1 - \mathbf{x}_0)\|_2^2\right] = \mathbb{E}_p\left[\left\|\mathbf{v}_\theta(\mathbf{x}_t, t) - \frac{\mathbf{x}_1 - \mathbf{x}_t}{1-t}\right\|_2^2\right]$$

$$= \frac{1}{(1-t)^2}\mathbb{E}_p\left[\|((1-t)\mathbf{v}_\theta(\mathbf{x}_t, t) + \mathbf{x}_t) - \mathbf{x}_1\|_2^2\right]. \tag{4}$$

Because the network $\mathbf{v}_\theta(\mathbf{x}_t, t)$ makes its predictions conditioned only on the intermediate state $\mathbf{x}_t$ and time $t$, the pointwise optimal predictor that minimizes this squared error is given by the conditional expectation. Therefore, the optimal value for $\mathbf{v}_\theta$ satisfies

$$((1-t)\mathbf{v}_\theta(\mathbf{x}_t, t) + \mathbf{x}_t) = \mathbb{E}_p[\mathbf{x}_1|\mathbf{x}_t].$$

Noting that the left-hand side only involves $(\mathbf{x}_t, t)$, we can define a new function $\mathbf{m}_\theta(\mathbf{x}_t, t)$ to replace the left-hand side. We then train by directly regressing $\mathbf{m}_\theta(\mathbf{x}_t, t)$ toward the data endpoint $\mathbf{x}_1$. The relationship between $\mathbf{m}_\theta$ and $\mathbf{v}_\theta$ is

$$\mathbf{m}_\theta = (1-t)\mathbf{v}_\theta(\mathbf{x}_t, t) + \mathbf{x}_t, \qquad \mathbf{v}_\theta = \frac{\mathbf{m}_\theta(\mathbf{x}_t, t) - \mathbf{x}_t}{1-t}. \tag{5}$$

**The SC-Flow method.** We introduce another neural function $\tilde{\mathbf{m}}_\theta$ to directly learn $\mathbf{x}_1$. The learning objective for $\tilde{\mathbf{m}}_\theta$ is

$$L_x(\theta) = \|\tilde{\mathbf{m}}_\theta(\mathbf{x}_t, t) - \mathbf{x}_1\|_2^2. \tag{6}$$

Here we use the same $\theta$ to denote the network parameters, since later both $\mathbf{v}_\theta$ and $\tilde{\mathbf{m}}_\theta$ are parameterized by the same network. Compared with Equation (4), this objective removes the leading factor $1/t^2$ to improve training stability; empirically, we find that this change has a negligible impact on performance.

Since $\mathbf{v}_\theta$ and $\tilde{\mathbf{m}}_\theta$ are computed from different neural functions, we need to make them consistent with each other. We include another consistency term in the training objective

$$L_c = \|\tilde{\mathbf{m}}_\theta(\mathbf{x}_t, t) - \mathbf{m}_\theta(\mathbf{x}_t, t)\|_2^2 = \|\tilde{\mathbf{m}}_\theta(\mathbf{x}_t, t) - ((1-t)\mathbf{v}_\theta(\mathbf{x}_t, t) + \mathbf{x}_t)\|_2^2. \tag{7}$$

The entire training objective is

$$\mathcal{L}(\theta) = w_v L_{\text{FM}}(\theta) + w_x L_x(\theta) + w_c L_c(\theta). \tag{8}$$

This joint training objective is illustrated in Figure 1.

**Parameterization.** While $\mathbf{v}_\theta$ and $\tilde{\mathbf{m}}_\theta$ play different roles, their required information is similar. In this work, we train a *single* network $\text{nn}_\theta$ to compute both functions as illustrated in Figure 1. In particular, we use a binary flag $b \in \{0, 1\}$ to indicate which function it computes

$$\mathbf{v}_\theta(\mathbf{x}_t, t) := \text{nn}_\theta(\mathbf{x}_t, t, b{=}0), \qquad \tilde{\mathbf{m}}_\theta(\mathbf{x}_t, t) := \text{nn}_\theta(\mathbf{x}_t, t, b{=}1). \tag{9}$$

From $\tilde{\mathbf{m}}_\theta$, we derive the velocity:

$$\tilde{\mathbf{v}}_\theta = \frac{\tilde{\mathbf{m}}_\theta(\mathbf{x}_t, t) - \mathbf{x}_t}{1 - t} \tag{10}$$

We refer to the original flow $\mathbf{v}_\theta$ as **V-Flow** and to our new flow $\tilde{\mathbf{v}}_\theta$ as **X-Flow**.

We also consider conditional generation. Let $y$ denote a conditioning variable (e.g., a class label). We augment the inputs with $y$, so the network becomes $\text{nn}_\theta(\mathbf{x}_t, t, b, y)$.

In the implementation, we embed the time $t$, the indicator $b$, and the condition $y$ with vectors and send their sum into the neural network, $e(t, y, b) = e_t(t) + e_y(y) + e_b(b)$. Here, we assume $y$ is a global, discrete conditioning variable (such as a class label) that can be mapped to a single embedding vector and added directly to the network's intermediate states.

As discussed in Section 2, diffusion-based and flow-based generative models share a few common design choices of network parameterizations. In this sense, our approach is related in spirit to the dual-output diffusion model of Benny & Wolf (2022). However, beyond the broader distinction between diffusion and flow frameworks, our method differs in two key aspects. First, SC-Flow introduces an explicit consistency objective that enforces agreement between the predicted velocity and data endpoint, whereas dual-output diffusion (Benny & Wolf, 2022) does not impose such a constraint between outputs. Second, our approach employs a single shared network with a lightweight toggle mechanism to produce both predictions, enabling tighter coupling between the two parameterizations with minimal additional complexity. As shown in our experiments, these design choices play an important role in improving both optimization stability and generation performance.

**Sampling.** In the generative procedure, we have two vector fields $\mathbf{v}_\theta(\mathbf{x}_t, t)$ and $\tilde{\mathbf{v}}_\theta(\mathbf{x}_t, t)$ for sampling. We can use either of them for sampling with path integration. We can also blend them in the sampling procedure.

$$\dot{\mathbf{x}}_t = \begin{cases} \tilde{\mathbf{v}}_\theta(\mathbf{x}_t, t), & t \leq \tau, \\ \mathbf{v}_\theta(\mathbf{x}_t, t), & t > \tau. \end{cases} \tag{11}$$

In our experiment, we use $\tau{=}0.5$. Abalation study can be found in Section A4. The blending strategy slightly improves generation performance in the experiment, though $\mathbf{v}_\theta$ and $\tilde{\mathbf{v}}_\theta$ do not have significant differences because of the consistent term. The strategy is justified by the stability during inference. Detailed analysis can be found in analysis section.

**Computation.** Compared with a baseline that predicts $\mathbf{v}_\theta$ with a single network, our model introduces only a minimal number of additional parameters –the embedding of the binary switch $b$. Training time increases only slightly because the computations for $\mathbf{v}_\theta$ and $\mathbf{m}_\theta$ are largely shared. At sampling time, the runtime is nearly identical to that of the baseline flow-matching model: the compute budget is essentially unchanged, and we simply toggle $b$ to select $\mathbf{v}_\theta$ or $\tilde{\mathbf{v}}_\theta$.

**Algorithm 1: SC-Flow training (single head)**

1: **Input:** $(w_v, w_x, w_c)$, clamp $\varepsilon$, optimizer
2: **for** minibatches $(x_0, x_1, y)$ in dataset **do**
3:      Draw $t \sim \mathcal{U}([\varepsilon,\, 1{-}\varepsilon])$
4:      $x_t \leftarrow (1{-}t)x_0 + tx_1, \quad u \leftarrow x_1 - x_0$
5:      $\mathbf{v}_\theta \leftarrow \text{nn}_\theta(x_t, t, y, b{=}0)$
6:      $\tilde{\mathbf{m}}_\theta \leftarrow \text{nn}_\theta(x_t, t, y, b{=}1)$
7:      Compute $L_{\text{FM}}, L_{\text{x}}, L_c$
8:      $\mathcal{L}(\theta) \leftarrow w_v L_{\text{FM}}(\theta) + w_x L_x(\theta) + w_c L_c(\theta)$
9:      Update $\theta$ with $\nabla_\theta L$
10: **end for**

**Algorithm 2: SC-Flow sampling (ODE)**

1: **Input:** prior $x_0 \sim p_0$, condition $y$, clamp $\varepsilon$
2: Choose per-step drift rule using equation 11
3: **for** $t$ from 0 to 1 with an ODE solver **do**
4:      $\mathbf{v}_\theta \leftarrow \text{nn}_\theta(x_t, t, y, b{=}0)$
5:      $\tilde{\mathbf{m}}_\theta \leftarrow \text{nn}_\theta(x_t, t, y, b{=}1)$
6:      $\tilde{\mathbf{v}}_\theta \leftarrow (\hat{x}_1 - x_t)/\max(1{-}t, \varepsilon)$
7:      $\dot{x}_t \leftarrow \text{mix}(\mathbf{v}_\theta, \tilde{\mathbf{v}}_\theta, t)$ as described in Equation (11)
8:      Advance $x_t$ one ODE step using $\dot{x}_t$
9: **end for**
10: **return** $x_{t=1}$

## 5 Analysis of SC-Flow

In this section, we analyze the learning errors associated with fitting the velocity versus the data endpoint. By examining the target variance, the dimensionality of the optimal vector fields, and the asymptotic error during sampling, we reveal a fundamental trade-off that motivates our dual-target SC-Flow architecture. We first demonstrate why endpoint prediction (X-Flow) provides superior training stability, and subsequently show why velocity prediction (V-Flow) is necessary to prevent inference instability near the data manifold.

We base our analysis on the following model assumption. We assume the independent coupling of noise and the data distribution, $p(\mathbf{x}_0, \mathbf{x}_1) = p(\mathbf{x}_0)p(\mathbf{x}_1)$. The conditional path is a linear interpolation $x_t = (1 - t)x_0 + tx_1$ for $t \in (0, 1)$. The analysis can be extended to the case of non-linear noise scheduling (Tsimpos et al., 2025), but we focus on the linear case for notational simplicity. The setting here represents practices in real applications (Liu et al., 2022; Esser et al., 2024).

**Variance and the intrinsic dimension of learning targets.** We first consider the learning targets of the neural network in the V-Flow and X. Given $(\mathbf{x}_t, t)$, let $p(\mathbf{x}_0, \mathbf{x}_1 | \mathbf{x}_t, t)$ be the conditional distribution of paths passing through $\mathbf{x}_t$. The conditional velocity is $\mathbf{u} = \frac{\mathbf{x}_1 - \mathbf{x}_t}{1 - t} = \mathbf{x}_1 - \mathbf{x}_0$. Then:

$$\text{Var}[\mathbf{x}_1 | \mathbf{x}_t, t] = (1 - t)^2 \text{Var}[\mathbf{u} | \mathbf{x}_t, t]. \tag{12}$$

We can see that $\mathbf{x}_1$ has a lower variance than $\mathbf{u}$. The finite-sample estimation of the true target $\mathbb{E}[\mathbf{x}_1 | \mathbf{x}_t, t]$ has lower variance than the velocity target $\mathbb{E}[\mathbf{u} | \mathbf{x}_t, t]$. For image data, noise in $\mathbf{u}$ degrades or even destroys the smoothness property of images, making patterns difficult to learn.

Li & He (2025) advocates for X-Flow by suggesting that data resides on a low-dimensional manifold and is thus inherently easier to fit, but their justification relies primarily on small-scale toy problems. We extend this intuition by providing a formal analysis. Consistent with our previous notation, we consider data $\mathbf{x}$ in a $d$-dimensional space. Following Zhang et al. (2025), we assume the data can be embedded into a lower-dimensional latent space $\mathbb{R}^l$ (where $l < d$). Specifically, we assume $\mathbf{x}$ can be perfectly reconstructed from its latent representation $\mathbf{z} \in \mathbb{R}^l$: $\mathbf{z} = Q\mathbf{x}, \mathbf{x} = Q\mathbf{z}$, with $Q \in \mathbb{R}^{d \times l}$ being a column-orthonormal matrix ($Q^\top Q = I_l$). Therefore, the latent space $\mathbb{R}^l$ contains the latent data manifold. Although lossless linear encoding is rarely achievable in complex real-world applications, it remains a robust analytical tool, as the primary variance of high-dimensional data is often concentrated within a few principal components.

The vector field on $\mathbf{x}$ entails a vector field on $\mathbf{z}$. Given the conditional flow $\mathbf{x}_t = t\mathbf{x}_1 + (1 - t)\mathbf{x}_0$, the projected conditional flow is $\mathbf{z}_t = t\mathbf{z}_1 + (1 - t)\mathbf{z}_0$, where $\mathbf{z}_0 = Q^\top \mathbf{x}_0$. If the optimal velocity estimator in the data and latent spaces are respectively $\mathbf{v}_h^*(\mathbf{x}_t) = \mathbb{E}[\mathbf{x}_1 - \mathbf{x}_0 | \mathbf{x}_t]$ and $\mathbf{v}_l^*(\mathbf{z}_t) = \mathbb{E}[\mathbf{z}_1 - \mathbf{z}_0 | \mathbf{z}_t]$. Because of the noise term $\mathbf{x}_0$ in the high-dimensional space, the optimal velocity $\mathbf{v}_h^*$ is still in the high-dimensional space. In particular, the two vector fields have the following relationship (Zhang et al., 2025). We put the proof in Appendix A.

$$\mathbf{v}_h^*(\mathbf{x}_t) = Q\mathbf{v}_l^*(Q^\top \mathbf{x}_t) - \frac{1}{1 - t}(I - QQ^\top)\mathbf{x}_t. \tag{13}$$

In contrast, with the assumption of linear construction, the predicting target of X-Flow stays in an $l$-dimensional subspace. Let $\mathbf{m}_h^*(\mathbf{x}_t) = \mathbb{E}[\mathbf{x}_1 | \mathbf{x}_t]$ and $\mathbf{m}_l^*(\mathbf{x}_t) = \mathbb{E}[\mathbf{z}_1 | \mathbf{z}_t]$ be the optimal predicting targets

of X-Flow, respectively, in the data and latent spaces. We can separate the linear components of $\mathbf{x}_t$ by $\mathbf{x}_t = (1-t)\mathbf{x}_0 + tQ\mathbf{z}_1 = (1-t)(I - QQ^\top)\mathbf{x}_0 + Q\mathbf{z}_t$. Let $\mathbf{s}_t = (1-t)(I - QQ^\top)\mathbf{x}_0$. With the setup of the learning problem: 1) the isotropic Gaussian distribution of $\mathbf{x}_0$ and 2) the independent coupling, $\mathbf{s}_t$ is independent of $\mathbf{z}_0 = Q^\top \mathbf{x}_0$ and $\mathbf{z}_1 = Q^\top \mathbf{x}_1$. Then we have the relationship between the two optimal predicting targets:

$$\mathbf{m}_h^*(\mathbf{x}_t) = \mathbb{E}[Q\mathbf{z}_1|\mathbf{x}_t] = \mathbb{E}[Q\mathbf{z}_1|\mathbf{z}_t, \mathbf{s}_t] = Q\mathbb{E}[\mathbf{z}_1|\mathbf{z}_t] = Q\mathbf{m}_l^*(\mathbf{z}_t). \tag{14}$$

This result indicates that the network in the X-Flow setup can predict targets in a low-dimensional latent space; therefore, it is more likely to reduce the learning error than the network that directly predicts velocity values. For these two reasons, we argue that it is easier to train the neural network in the X-Flow setting.

**Inference stability near the data manifold.** Despite its training stability, deriving the velocity from an endpoint predictor introduces severe inference instability as the trajectory approaches the data manifold. Specifically, the learning error of $\tilde{\mathbf{m}}_\theta$ will be scaled up by $\frac{1}{1-t}$. With the conditional distribution $p(\mathbf{x}_1, \mathbf{x}_0|\mathbf{x}_t, t)$, let $\delta = \|\mathbf{v}_\theta(\mathbf{x}_t, t) - \mathbb{E}[\mathbf{u}]\|_2^2$ be the learning error of the V-Flow model. Let $\epsilon = \|\tilde{\mathbf{m}}_\theta(\mathbf{x}_t, t) - \mathbb{E}[\mathbf{x}_1]\|_2^2$ be the learning error of the X-Flow model, then the error of the velocity $\tilde{\mathbf{v}}_\theta$ computed from $\tilde{\mathbf{m}}_\theta$ is $\tilde{\delta} = \frac{\epsilon}{(1-t)^2}$, which is scaled up because of the factor $\frac{1}{1-t}$.

At the early stage of the path, when $(1-t)$ is significantly greater than 0, it is hard to argue whether $\mathbf{v}_\theta$ or $\tilde{\mathbf{v}}_\theta$ has smaller error for reasons from both sides. However, the analysis does indicate that X-Flow has an advantage when the data dimension is much higher the compact latent dimension. The previous work by Li & He (2025) studies parameterization methods in the pixel space and shows that some V-Flow networks $\mathbf{v}_\theta$ cannot even converge while the X-Flow model $\tilde{\mathbf{m}}_\theta$ can. The finding is consistent with our theoretical analysis above. However, at the late stage when $1 - t$ is small, and the error $\epsilon$ of the X-Flow model is significantly amplified by the factor $\frac{1}{1-t}$, then X-Flow is a worse choice than V-Flow.

We can compare the learning errors on a validation set by checking their learning objectives. The two errors $\delta$ and $\tilde{\delta}$ can be compared by:

$$\delta = \mathbb{E}_{\mathbf{x}_t}[\|(\mathbf{v}_\theta - \mathbb{E}[\mathbf{u}|\mathbf{x}_t])\|_2^2] = \mathbb{E}[\|\mathbf{v}_\theta - \mathbf{u}\|_2^2] - \mathbb{E}_{\mathbf{x}_t}[\mathrm{tr}(\mathrm{Var}(\mathbf{u}))], \tag{15}$$

$$\tilde{\delta} = \mathbb{E}_{\mathbf{x}_t}[\|(\tilde{\mathbf{v}}_\theta - \mathbb{E}[\mathbf{u}|\mathbf{x}_t])\|_2^2] = \mathbb{E}[\|\tilde{\mathbf{v}}_\theta - \mathbf{u}\|_2^2] - \mathbb{E}_{\mathbf{x}_t}[\mathrm{tr}(\mathrm{Var}(\mathbf{u}))]. \tag{16}$$

The second term is a constant and does not affect the comparison. Our experiment later shows that $\tilde{\delta}$ is significantly larger than $\delta$ when $t$ approaches 1.

# 6 Experiments

We empirically evaluate SC-Flow on image generation tasks. We first check the performance of SC-Flow and then examine the effect of the proposed training method with a series of ablation studies.

**Datasets and evaluation metrics.** The experiment is conducted on the two well-known datasets, CIFAR-10 (Krizhevsky, 2009) and ImageNet $256 \times 256$ (Deng et al., 2009). The evaluation metrics are Fréchet Inception Distance (FID) (Heusel et al., 2017), sFID (Siarohin et al., 2019), Inception Score (IS) (Salimans et al., 2016), and Precision (Kynkäänniemi et al., 2019). Lower FID/sFID and higher IS and Precision correspond to better performance.

Our primary baseline for performance comparison is Rectified Flow. All methods share the same neural backbone architecture, with the only difference being the additional switch $b$ introduced in SC-Flow. SC-Flow supports three sampling strategies: V-Flow, X-Flow, and a mixed approach defined in Equation (11). We denote the three strategies with suffixes -V, -X, and -Mix, respectively.

## 6.1 Evaluation with CIFAR-10

For our CIFAR-10 experiments, we use a U-Net backbone with an architecture similar to the one used in DDPM++ (Nichol & Dhariwal, 2021), which is a standard for this dataset. Both the baseline and SC-Flow models are trained for 250K iterations. For evaluation, we generate samples using Heun's method with 250 steps to ensure stable ODE integrations.

Table 1: Main results on CIFAR-10 (250K iterations, 250 sampling steps). SC-Flow consistently outperforms the Rectified Flow baseline.

| Model | FID↓ | sFID↓ | IS↑ | Precision↑ |
|---|---|---|---|---|
| RectifiedFlow | 3.12 | 3.34 | 9.47 | 0.72 |
| SC-Flow-V | 2.49 | 2.93 | 9.61 | 0.73 |
| SC-Flow-X | **2.41** | 2.88 | 9.64 | 0.73 |
| SC-Flow-Mix ($\tau$=0.5) | **2.41** | **2.83** | **10.65** | 0.73 |

As shown in Table 1, SC-Flow demonstrates a significant performance improvement over the standard rectified flow baseline on CIFAR-10. The SC-Flow-Mix and SC-Flow-X variants achieve the best FID score of **2.41**, a substantial 22.8% relative improvement over the baseline's 3.12. All SC-Flow sampling modes achieve better performance than the baseline method, indicating that the new training scheme also improves the V-flow $\mathbf{v}_\theta$. This strong performance with a convolutional U-Net architecture (Ronneberger et al., 2015) (as opposed to a Transformer (Vaswani et al., 2017)) highlights the general applicability and benefit of our proposed method.

## 6.2 Evaluation with ImageNet 256x256

Our experiments on ImageNet use the DiT-L/4 and DiT-XL/2 (Peebles & Xie, 2023) backbones as the foundation for our models. Here L and XL are mode sizes defined in the original paper. The number are patch sizes used by the generative model. We use "-L" and "-XL" to indciate these two settings.

Our standard Rectified Flow baselines are trained following the SiT (Ma et al., 2024). The SC-Flow model employs the same backbone architecture, except for the input of the binary switch $b$. The L/4 models are trained for 400K iterations and the XL/2 models for 800K iterations, both with a constant learning rate of $1 \times 10^{-4}$. For evaluation, all models are sampled using the Dopri5 ODE solver (Dormand & Prince, 1980) with 250 steps.

In this experiment, we include DiT as a baseline. DiT uses the same network backbone but is a diffusion-based generative method. We also train a flow matching model with only X-flow: we train $\tilde{\mathbf{m}}_\theta$ with the objective $L_\mathrm{x}$ and generate images with the derived X-flow $\tilde{\mathbf{v}}_\theta$. We denote this method as RectifiedFlow-L-X or RectifiedFlow-XL-X. All experiments are conducted with classifier free guidance (cfg) (Ho & Salimans, 2022) during sampling.

Table 2 presents our main results on ImageNet. SC-Flow substantially outperforms the Rectified Flow baselines across all metrics. On the L/4 model, our mixed-inference strategy (SC-Flow-L-Mix) improves the FID score of Rectified Flow from 11.53 to **9.85**, a relative improvement of 14.6%. The gains are even more pronounced on the larger DiT-XL/2 model, where SC-Flow-XL-Mix reduces the FID from 6.22 to **4.19**, a 32.6% relative improvement. Notably, SC-Flow also achieves a significantly higher Inception Score (IS) and Precision, indicating that the generated samples are not only more realistic but also more faithful to the training data. SC-Flow using the other two sampling methods also consistently outperforms the baseline, with the X-flow performing slightly better than the V-flow.

**Training Efficiency.** To evaluate the impact of dual-target supervision on optimization, we compare the training convergence of SC-Flow against standard SiT and DiT baselines. As illustrated in Figure 2, SC-Flow exhibits a markedly superior learning curve. By supervising both the local velocity and the global destination, the model benefits from the variance reduction properties established in our theoretical analysis. Specifically, SC-Flow reaches an FID of 2.0 in approximately 2M iterations, whereas the SiT baseline requires nearly 7M iterations to achieve comparable quality. This 3.5× acceleration in training efficiency highlights that the consistency loss stabilizes the gradient flow, allowing the network to capture the underlying data manifold more effectively in the early stages of training.

Table 2: Performances on ImageNet $256 \times 256$. SC-Flow with all three sampling methods has significant improvements over the baseline model.

(a) L/4 (cfg=4.0, 400K training steps, 250 sampling steps)

| Model | Training Steps | FID↓ | sFID↓ | IS↑ | Precision↑ |
|---|---|---|---|---|---|
| DiT-L | 400K | 11.84 | 12.22 | 116.50 | 0.63 |
| SiT-L | 400K | 11.53 | 12.06 | 110.75 | 0.65 |
| SiT-L-X | 400K | 11.48 | 12.02 | 112.63 | 0.65 |
| SC-Flow-L-V | 400K | 9.92 | 10.04 | 143.09 | **0.68** |
| SC-Flow-L-X | 400K | 10.63 | 11.90 | 137.86 | 0.67 |
| SC-Flow-L-Mix ($\tau$=0.5) | 400K | **9.85** | **10.02** | **145.61** | **0.68** |

(b) XL/2 (cfg=1.5, 800K training steps, 250 sampling steps)

| Model | Training Steps | FID↓ | sFID↓ | IS↑ | Precision↑ |
|---|---|---|---|---|---|
| DiT-XL | 800K | 6.41 | 6.67 | 175.01 | 0.73 |
| DiT-XL | 7M | 2.27 | 4.60 | 278.24 | 0.83 |
| SiT-XL | 800K | 6.22 | 6.65 | 157.46 | 0.73 |
| SiT-XL-X | 800K | 6.19 | 6.72 | 155.25 | 0.74 |
| SiT-XL | 7M | 2.06 | 4.60 | 258.09 | 0.81 |
| SC-Flow-XL-V | 800K | 4.21 | 4.68 | 184.97 | **0.79** |
| SC-Flow-XL-X | 800K | 4.28 | 4.68 | 183.48 | **0.79** |
| SC-Flow-XL-Mix ($\tau$=0.5) | 800K | **4.19** | **4.65** | **185.01** | **0.79** |
| SC-Flow-XL-Mix ($\tau$=0.5) | 4M | **1.86** | **4.21** | **285.93** | **0.85** |

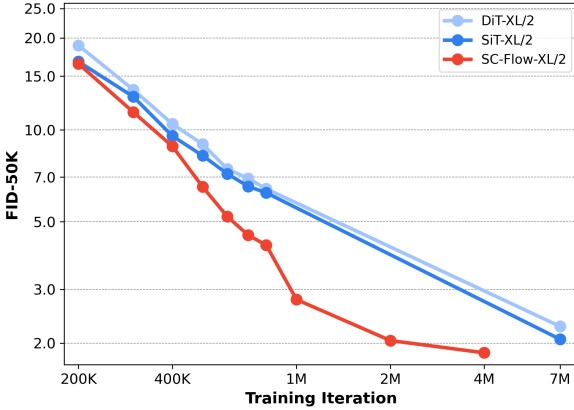

Figure 2: **Training Convergence.** FID-50K vs. training iterations on ImageNet. SC-Flow achieves significantly faster convergence, reaching the performance of the SiT baseline in $3.5\times$ fewer iterations.

### 6.3 Ablation Studies.

To validate our design choices and understand how SC-Flow improves the generation performance, we conduct a series of ablation studies focusing on the role of the consistency loss ($L_c$).

**The X-flow or the V-flow does not work well separately.** To isolate the effect of the X-flow training from minimizing $L_{\mathrm{x}}(\theta) = \|\tilde{\mathbf{m}}_\theta(\mathbf{x}_t, t) - \mathbf{x}_1\|_2^2$, we can train a Rectified Flow model by minimizing $L_{\mathrm{x}}(\theta)$ only and then draw samples from $\tilde{\mathbf{v}}_\theta$ derived from $\tilde{\mathbf{m}}_\theta$. As we mentioned before, this method is labeled as RectifiedFlow-X, and its performance is shown in Table 2. Its performance is nearly identical to the performance of the baseline Rectified Flow. Note that the V-flow $\mathbf{v}_\theta$ is trained against a corrupted image

(a) The scale of $L_c$ versus sampling steps.

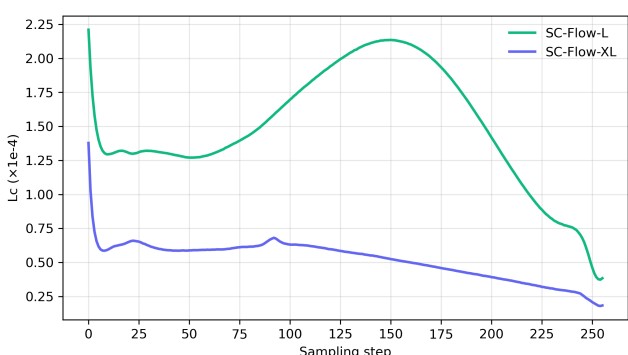

(b) Measurement of path straightness.

| Model | Max-dev$\downarrow$ | Mean-cos$\uparrow$ |
|---|---|---|
| RectifiedFlow-L | 0.256 | 0.828 |
| SC-Flow-L-V | 0.250 | 0.833 |
| SC-Flow-L-X | 0.248 | 0.833 |

(c) FID versus consistency weight $w_c$.

| $w_c$ | **0.0** | **0.1** | **0.2** | **0.3** |
|---|---|---|---|---|
| SC-Flow-L-Mix | 11.14 | 9.85 | 9.86 | 9.91 |

Figure 3: Analysis of the effect of the consistency term in SC-Flow. **(a)** The consistency loss $L_c$ measured during a 250-step ODE solver. The gap between $\mathbf{m}_\theta$ and $\tilde{\mathbf{m}}_\theta$ is largest at early steps and converges to zero. **(b)** The evaluation of path straightness shows SC-Flow generates more direct paths than Rectified Flow. **(c)** A sweep over the consistency weight $w_c$ shows that $w_c > 0$ is critical.

$\mathbf{x}_1 - \mathbf{x}_0$, while the X-flow is trained against a clean image $\mathbf{x}_1$. This result demonstrates that the different training targets do not bring a clear performance difference.

**The consistency loss is necessary.** In Figure 3(c), we vary the the weight $w_c$ of the consistency term $L_c(\theta)$. The best performance is achieved with $w_c \in \{0.1, 0.2\}$. When $w_c = 0$ eliminates the consistency constraint, SC-Flow only has a slight performance improvement over the baseline Rectified Flow (11.14 vs. 11.53 FID). We hypothesize that the improvement is from the network sharing, which is the focus of the next experiment. This result reveals that the consistency loss $L_c$ is essential for unlocking the full potential of our method. Our qualitative study later in Figure 6 also shows that the consistency loss improves the quality of the generated images.

**The shared neural architecture brings performance gain.** Sharing the same neural architecture clearly saves computation. At the same time, they learn targets that are quite related, so sharing the architecture should also be a reasonable choice for performance consideration. To verify this hypothesis, we train two independent DiT-L models for the V-flow and the X-flow. The training objective is the same as SC-Flow models above. We denote this new setup with the suffix "-2net". The performance of the two models under this setup is shown in Table 3. The performance still improves upon the baseline Flow Matching model. The improvement is solely due to the consistency term because that's the only difference from the baseline. However, their performance cannot match that of SC-Flow models with shared networks, which indicates the performance benefit of learning related targets with a single network backbone.

Table 3: The performance of SC-Flow models using two separate networks for the V-Flow and X-Flow.

| Model | FID$\downarrow$ | sFID$\downarrow$ |
|---|---|---|
| SC-Flow-L-V-2net | 10.92 | 12.15 |
| SC-Flow-L-X-2net | 10.57 | 10.77 |

**Consistency helps to overcome randomness in training.** As we analyzed in Section 3, the V-flow and the X-flow should be equivalent when there are infinite training examples, and the consistency term would not be useful. We hypothesize that the consistency between the V-flow and the X-flow is very beneficial when the model is trained with limited data and noise. To verify this hypothesis, we increase the amount of random noise examples to 4 times the training samples in *each* training batch, and therefore reduce the random noise in the training process.

Table 4: Performance of models trained with the 1:4 ratio of training samples and noise samples on CIFAR-10.

| Model | FID$\downarrow$ | sFID$\downarrow$ |
|---|---|---|
| RectifiedFlow (1:4) | 2.57 | 2.94 |
| SC-Flow-Mix (1:4) | 2.28 | 2.73 |

We conducted this experiment with the CIFAR-10 dataset. The performance of the trained models is shown in Table 4. The new training strategy improves the performance of both models because it essentially increases the training batches of the both models. However, this new setup shows less performance improvement with SC-Flow than the standard setup: the reduction of FID is (2.57 - 2.28) from the previous reduction (3.12 - 2.41). This provides strong evidence supporting our hypothesis.

**SC-Fow slightly straightens sampling paths**   We compare SC-Flow and Rectified Flow in terms of the straightness of sampling paths. The straightness is measured by Max-dev (the maximum deviation from the chord connecting the two ends of the path) and Mean-cos (average cosine similarity between each step and the chord). Lower Max-dev or higher Mean-cos indicates more straight paths. The results are shown in Figure 3(b). SC-Flow achieves slightly straighter paths. One possible explanation is that the X-flow aims to predict the destination and thus has a better chance to learn straight paths. We also plot the consistency loss $L_c$ in Figure 3(a) at each step of the ODE path. For both L and XL models, the difference is very small (at the scale of 1e-4), although the two models share a similar pattern. The inconsistency is highest near $t = 0$ where the path is most uncertain, and converges towards zero as $t \to 1$.

(a) FID-50K vs. Training Steps. Left: L Models. Right: XL Models.

(b) Training and Sampling Time Comparison (seconds) on ImageNet

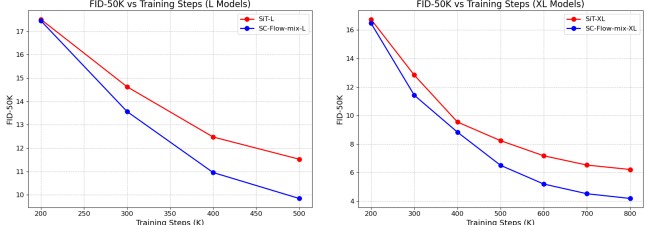

| Model | Tr./epoch | Spl./step |
|-------|-----------|-----------|
| SiT-L | 0.80 | 0.09 |
| SC-Flow-V-L | 1.01 | 0.10 |
| SiT-XL | 0.78 | 0.51 |
| SC-Flow-V-XL | 0.81 | 0.56 |

Figure 4: Efficiency and Training Dynamics. **(a)** SC-Flow has lower FID score than the baseline at every checkpoint. **(b)** The training overhead is negligible and sampling cost is identical.

**Computation efficiency.**   Figure 4 demonstrates that SC-Flow's significant performance gains are achieved with high efficiency. The plots on the left show the FID-50K score as a function of training steps for both L and XL models. At every evaluation checkpoint, SC-Flow maintains a consistent and substantial advantage over the standard Rectified Flow baseline, indicating not only a better final result but also faster convergence. The table on the right quantifies the computational cost. The training time per epoch for SC-Flow is only marginally higher than the baseline. This is due to our efficient single-head design, which requires only one additional forward pass through the shared network backbone to compute both targets. Crucially, the sampling time per step is nearly identical, as both methods require just one model evaluation per ODE step. Taken together, these results confirm that SC-Flow is a "plug-and-play" upgrade, offering superior sample quality and faster training convergence with a negligible increase in training cost and no change to the sampling budget.

**Stability during inference.**   To empirically validate the asymptotic behavior described in Section 5, we analyze the precision of the learned vector fields near the data manifold. This analysis demonstrates how different supervision signals affect the numerical stability of the ODE solver.

We evaluate the velocity prediction error using 2,000 images randomly sampled from the ImageNet validation set. From these 2,000 validation samples, we compute the learning objective $\|\mathbf{v}_\theta(\mathbf{x}_t, t) - (\mathbf{x}_1 - \mathbf{x}_0)\|_0^2$ with $\mathbf{v}$-s derived from different models: $\mathbf{v}_\theta$ from SC-Flow with $b = 0$ (SC-Flow-V), $\tilde{\mathbf{v}}_\theta$ derived from SC-Flow with $b = 1$ (SC-Flow-X), $\mathbf{v}_\theta$ from the baseline trained with velocity (Single-V), and $\tilde{\mathbf{v}}_\theta$ derived from the baseline trained with endpoints (Single-X). As shown in Figure 5, the empirical results show that X-Flow and V-Flow are not very distinguishable at the first stage of the path. When $t > 0.8$, the error of X-Flow begins to increase quickly because of the amplification effect in our discussion. SC-Flow clearly decreases the learning error throughout all time $t$. Even for SC-Flow, X-Flow still tends to have a large error at the end of the sampling

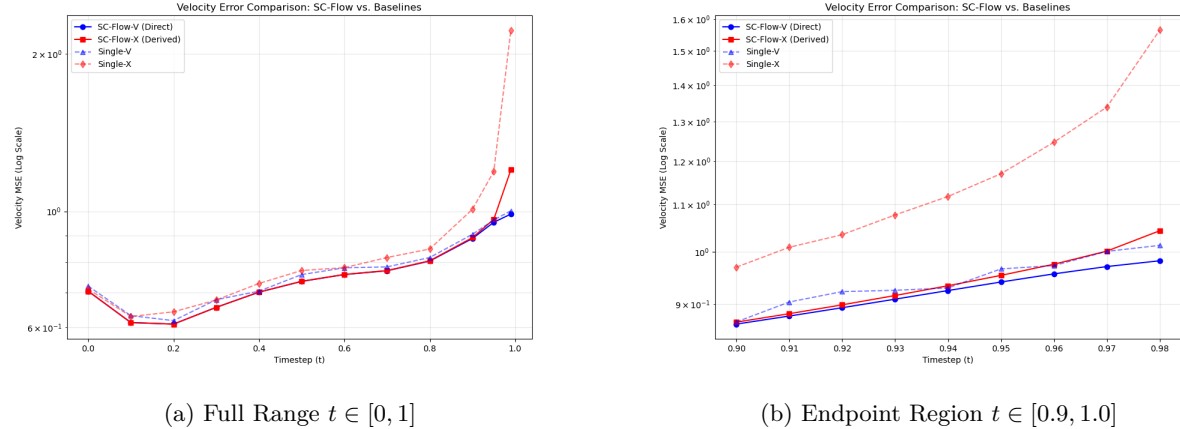

(a) Full Range $t \in [0, 1]$        (b) Endpoint Region $t \in [0.9, 1.0]$

Figure 5: **Velocity Error Comparison.** We compare the Mean Squared Error (MSE) of velocity predictions against ground truth for SC-Flow and single-target baselines. As predicted by inference stability analysis in Section 5, the derived velocity from endpoint predictors (red lines) spikes exponentially as $t \to 1$ due to the $(1-t)^{-1}$ error magnification. SC-Flow (blue solid) maintains the lowest overall error, demonstrating how joint supervision and consistency regularize the vector field against asymptotic instability. The right panel provides a focused view of this behavior near the data manifold.

stage. Our mixed sampling strategy avoids such error by using V-Flow in the latter part of the sampling procedure.

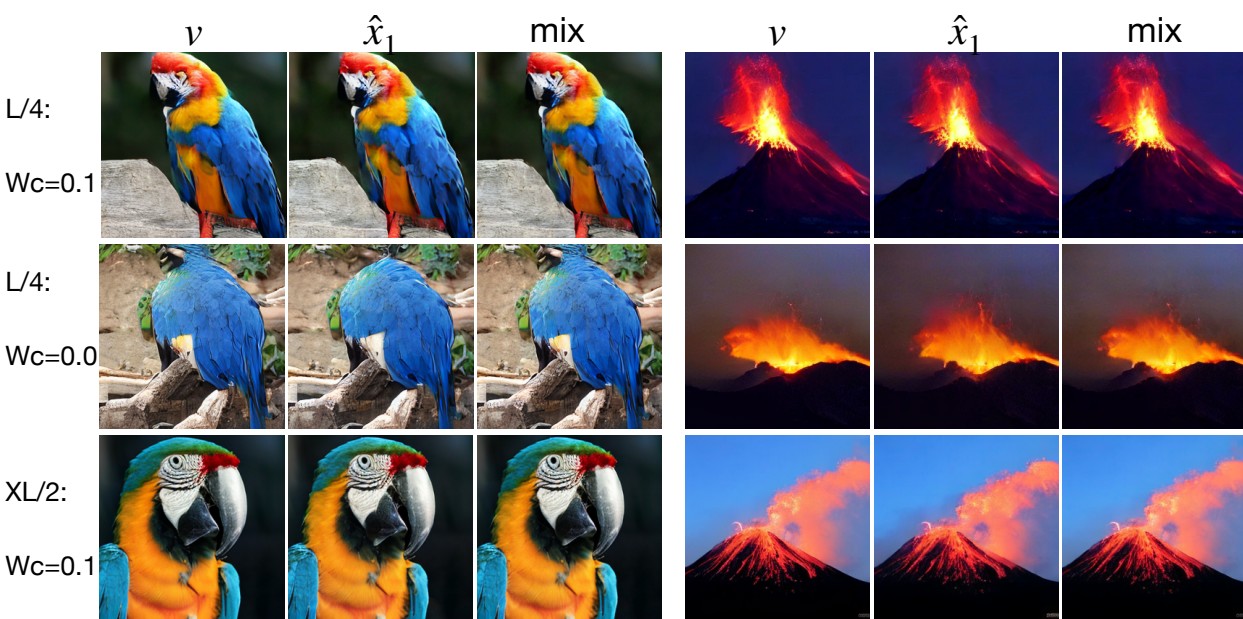

Figure 6: **Qualitative Comparison on ImageNet** $256 \times 256$**.** We compare samples from the L/4 and XL/2 SC-Flow models using V-Flow $\mathbf{v}_\theta$, X-Flow $\tilde{\mathbf{v}}_\theta$, and mixed inference. The bottom row (XL/2) shows higher fidelity than the L/4 model. The middle row ablates our consistency loss ($w_c = 0.0$), revealing a clear visual divergence between the velocity and endpoint predictions. The top row ($w_c = 0.1$) demonstrates that our proposed consistency loss forces the two predictions into alignment, producing more coherent and higher-quality samples.

**Qualitative Analysis.** Figure 6 provides a qualitative analysis of SC-Flow. As expected, the larger XL/2 model (bottom row) generates samples with noticeably higher fidelity and finer detail than the L/4 model (top row), consistent with the quantitative metrics. The critical comparison is between the model trained with our consistency loss ($w_c = 0.1$, top row) and the ablation without it ($w_c = 0.0$, middle row). When $w_c = 0.0$, the model's two predictions are unconstrained and learn inconsistent vector fields. This is visually apparent: the samples generated respectively from the V-flow and the X-flow exhibit clear discrepancies in texture, lighting, and fine details. For example, the parrot's perch has noticeable artifacts in the V-flow sample that are absent in the X-flow. By introducing the consistency loss, these two predictions are forced into agreement. The resulting images from both flows become nearly identical and have reduced artifacts. This visually confirms the significance of the consistent loss, which enforces the entire model to learn a coherent generation flow.

## 7 Limitations and Conclusion

**Limitations and future work.** While SC-Flow demonstrates strong empirical advantages, our current formulation presents a few limitations that offer avenues for future work.

First, our training objective and consistency loss are specifically formulated for linear rectified paths ($x_t = (1 - t)x_0 + tx_1$) and ODE-based sampling. Extending SC-Flow to non-linear trajectories, such as variance-preserving diffusion paths, would require careful derivation of the consistency constraint in a different way.

Second, SC-Flow needs to be validated on generation tasks with much higher resolutions and other modalities (e.g., text-to-image or video). As ambient dimensionality grows significantly larger than the intrinsic data manifold, joint optimization may require adaptive loss balancing to prevent the high-variance velocity targets from overpowering the shared network.

Third, enforcing the consistency constraint slightly increases training time. Although the network architecture is shared, computing both the velocity and endpoint predictions for the $L_c$ term requires an additional forward evaluation during each training step.

**Conclusion.** In this work, we theoretically and empirically investigate the distinct behaviors of different prediction targets within the rectified flow framework. We reveal a fundamental trade-off: predicting the data endpoint provides a low-variance, on-manifold signal that stabilizes training, while predicting the instantaneous velocity ensures bounded integration error for stable sampling near the data manifold. To unify these complementary strengths, we introduced **SC-Flow** (Self-Consistent Flow). By employing a lightweight algebraic consistency penalty, SC-Flow trains a single network with a binary control bit to concurrently predict both the local velocity and the global destination. At inference, this dual-target parameterization enables a zero-overhead switching policy that seamlessly transitions from X-Flow to V-Flow. Extensive experiments on CIFAR-10 and ImageNet 256×256 demonstrate that explicitly coupling local motion with global awareness significantly stabilizes optimization, straightens generation paths, and improves overall sample quality. Ultimately, SC-Flow offers a principled, plug-and-play approach to building stronger continuous-time generative models without requiring architectural upheaval or additional sampling compute.

## Acknowledgment

The authors thank the reviewers for their insightful comments and constructive feedback, which have significantly contributed to the improvement of this work. Jiajing Hu and Li-Ping Liu's work was supported by the U.S. National Science Foundation under Award No. 2239869.

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

# A   Appendix

## Appendix A.1   Proof

*Proof.* Let $P = QQ^\top$ denote the orthogonal projection matrix onto the data manifold $\mathcal{M}$, and $P^\perp = I - QQ^\top$ denote the projection onto its orthogonal complement. We can decompose the optimal ambient velocity target into its intrinsic and orthogonal components:

$$\mathbf{x}_1 - \mathbf{x}_0 = P(\mathbf{x}_1 - \mathbf{x}_0) + P^\perp(\mathbf{x}_1 - \mathbf{x}_0). \tag{17}$$

First, we analyze the orthogonal component. Because the data $\mathbf{x}_1 = Q\mathbf{z}_1$ resides strictly on the manifold, it has no orthogonal component ($P^\perp\mathbf{x}_1 = 0$). Applying $P^\perp$ to both sides of the flow trajectory $\mathbf{x}_t = t\mathbf{x}_1 + (1-t)\mathbf{x}_0$ yields:

$$P^\perp\mathbf{x}_t = (1-t)P^\perp\mathbf{x}_0 \quad \Longrightarrow \quad P^\perp\mathbf{x}_0 = \frac{1}{1-t}P^\perp\mathbf{x}_t. \tag{18}$$

Consequently, the orthogonal component of the velocity target is deterministic given $\mathbf{x}_t$:

$$\mathbb{E}[P^\perp(\mathbf{x}_1 - \mathbf{x}_0)|\mathbf{x}_t] = -\frac{1}{1-t}P^\perp\mathbf{x}_t = -\frac{1}{1-t}(I - QQ^\top)\mathbf{x}_t. \tag{19}$$

Next, we evaluate the intrinsic manifold component. Using $\mathbf{x}_1 = Q\mathbf{z}_1$ and the definition of the projected noise $\mathbf{z}_0 = Q^\top\mathbf{x}_0$, we have:

$$P(\mathbf{x}_1 - \mathbf{x}_0) = Q(Q^\top\mathbf{x}_1 - Q^\top\mathbf{x}_0) = Q(\mathbf{z}_1 - \mathbf{z}_0). \tag{20}$$

Assuming the ambient noise $\mathbf{x}_0$ is isotropic Gaussian, its orthogonal projection $P^\perp\mathbf{x}_0$ and intrinsic projection $Q^\top\mathbf{x}_0$ are statistically independent. Therefore, conditioning on the full ambient state $\mathbf{x}_t$ for the intrinsic variables is mathematically equivalent to conditioning strictly on the projected intrinsic state $\mathbf{z}_t = Q^\top\mathbf{x}_t$. Taking the conditional expectation yields:

$$\mathbb{E}[P(\mathbf{x}_1 - \mathbf{x}_0)|\mathbf{x}_t] = Q\mathbb{E}[\mathbf{z}_1 - \mathbf{z}_0|\mathbf{z}_t] = Q\mathbf{v}_l^*(\mathbf{z}_t). \tag{21}$$

Finally, by the linearity of expectation, summing the intrinsic and orthogonal conditional expectations completes the proof:

$$\begin{aligned}\mathbf{v}_h^*(\mathbf{x}_t) &= \mathbb{E}[P(\mathbf{x}_1 - \mathbf{x}_0)|\mathbf{x}_t] + \mathbb{E}[P^\perp(\mathbf{x}_1 - \mathbf{x}_0)|\mathbf{x}_t] \\ &= Q\mathbf{v}_l^*(Q^\top\mathbf{x}_t) - \frac{1}{1-t}(I - QQ^\top)\mathbf{x}_t.\end{aligned} \tag{22}$$

$\square$

## Appendix A.2   ImageNet Experiment Details

This section documents the full training and evaluation setup used for ImageNet $256\times256$ with DiT-L/4 and DiT-XL/2 backbones. Unless stated, SC-Flow and the Rectified-Flow baselines share identical settings.

**Data and Latent Encoding**

**Dataset and preprocessing.**   We train on ImageNet (train split) with class-conditional labels. Images are center-cropped and resized to $256 \times 256$, followed by random horizontal flip and normalization to $[-1, 1]$.

**Latent space (VAE).**   All models operate in the latent space of a frozen, pretrained VAE encoder (no finetuning). We use `diffusers AutoencoderKL.from_pretrained("stabilityai/sd-vae-ft-ema")`. Latents are scaled by the standard factor 0.18215 (as in Stable Diffusion).

Table 5: Latent-space and encoder specifics.

| Item | Encoder | Trainable | Image Size | Downsample | Latent Size | Latent Channels |
|------|---------|-----------|------------|------------|-------------|-----------------|
| VAE | SD-VAE-FT-{ema\|mse} | No (frozen) | 256×256 | ×8 | 32×32 | 4 |

## Model, Conditioning, and Mode Bit

**Backbones.** We use DiT-L/4 and DiT-XL/2 transformers (patch sizes 4 and 2 respectively). Exact widths/depths follow DiT; we summarize placeholders below for completeness.

Table 6: Backbone summary (filled from the provided model definitions).

| Model | Input Size | In Ch. | Patch | Tokens | Depth | Hidden | Heads | PosEnc |
|-------|-----------|--------|-------|--------|-------|--------|-------|--------|
| DiT-L/4 | 32 | 4 | 4×4 | 8×8=64 | 24 | 1024 | 16 | 2D sin/cos (frozen) |
| DiT-XL/2 | 32 | 4 | 2×2 | 16×16=256 | 28 | 1152 | 16 | 2D sin/cos (frozen) |

**Conditioning streams and SC-Flow mode.** We sum three embeddings into the adaLN-Zero conditioning vector at every block:

$$e(t, y, b) \;=\; E_t(t) \;+\; E_y(y) \;+\; E_b(b).$$

Time uses a sinusoidal projection (`frequency_embedding_size`= 256) followed by an MLP to the hidden size. Class labels use an embedding table of size `num_classes + 1` when classifier-free guidance (CFG) dropout is enabled. The SC-Flow *mode bit* $b \in \{0, 1\}$ selects which prediction the single head should output (local motion when $b{=}0$, global endpoint when $b{=}1$).

Table 7: Embedding dimensions and fusion.

| Stream | Source | Dim | Notes | Fusion |
|--------|--------|-----|-------|--------|
| Time | sinusoidal → MLP | $d_{\text{hid}}$ | freq emb size 256 | |
| Class | embedding table | $d_{\text{hid}}$ | (`num_classes`+1) if CFG | $E_t + E_y + E_b$ |
| Mode bit $b$ | embedding table (2 entries) | $d_{\text{hid}}$ | $b{=}0$: local motion; $b{=}1$: endpoint | |

## Optimization and Training Schedule

We train with AdamW (constant learning rate $1{\times}10^{-4}$, no weight decay) and exponential moving average (EMA) of weights. TensorFloat-32 (TF32) is enabled on A100/H200.

## Sampling and Evaluation

We integrate the probability-flow ODE using Dopri5 with a fixed budget of 250 steps. For SC-Flow we keep compute parity by evaluating only the active branch per step (one forward pass per step). We report FID-50K, sFID, IS, and Precision.

## Path Straightness Metrics: `mean_cos` and `max-dev`

**Setup and notation.** For a batch of discrete sampling paths $\{x_k^{(b)}\}_{k=0}^{S}$ with $b = 1, \ldots, B$ and $x_k^{(b)} \in \mathbb{R}^D$ (flattened images), define the *chord*

$$c^{(b)} \;=\; x_S^{(b)} - x_0^{(b)}, \qquad u^{(b)} \;=\; \frac{c^{(b)}}{\|c^{(b)}\|} \;\; \text{(unit chord)}. \tag{23}$$

Let the *step vector* be

$$s_k^{(b)} \;=\; x_{k+1}^{(b)} - x_k^{(b)}, \qquad k = 0, \ldots, S-1. \tag{24}$$

Table 8: Optimizer, precision, and regularization.

| Setting | Optimizer | LR | Weight Decay | Betas | Eps | EMA Decay | AMP |
|---------|-----------|-----|--------------|-------|-----|-----------|-----|
| All | AdamW | $1\times10^{-4}$ | 0 | (0.9, 0.999) | $1\times10^{-8}$ | 0.9999 | `False` |

Table 9: Sampling configuration and guidance.

| Model | Solver | Steps | Switch $\tau$ | CFG Scale |
|-------|--------|-------|---------------|-----------|
| DiT-L/4 | Dopri5 | 250 | 0.5 | 4.0 |
| DiT-XL/2 | Dopri5 | 250 | 0.5 | 1.5 |

**Mean cosine to the chord (`mean_cos`).** The per–time-step cosine between the step and the chord direction is

$$\gamma_k^{(b)} \;=\; \frac{\langle s_k^{(b)}, \, u^{(b)} \rangle}{\|s_k^{(b)}\|} \;=\; \frac{\langle x_{k+1}^{(b)} - x_k^{(b)}, \, \frac{x_S^{(b)} - x_0^{(b)}}{\|x_S^{(b)} - x_0^{(b)}\|} \rangle}{\|x_{k+1}^{(b)} - x_k^{(b)}\|}. \tag{25}$$

We average over steps and the batch:

$$\texttt{mean\_cos} \;=\; \frac{1}{BS} \sum_{b=1}^{B} \sum_{k=0}^{S-1} \gamma_k^{(b)}. \tag{26}$$

Values closer to 1 indicate that steps remain well aligned with the global chord (straighter paths).

**Maximum lateral deviation (`max-dev`).** For each intermediate point, decompose $x_k^{(b)}$ into chord-parallel progress and orthogonal residual. Define the signed progress along the chord,

$$d_k^{(b)} \;=\; \langle x_k^{(b)} - x_0^{(b)}, \, u^{(b)} \rangle, \tag{27}$$

the orthogonal projection onto the chord,

$$p_k^{(b)} \;=\; x_0^{(b)} + d_k^{(b)} \, u^{(b)}, \tag{28}$$

and the residual (lateral) displacement

$$r_k^{(b)} \;=\; x_k^{(b)} - p_k^{(b)}, \qquad \delta_k^{(b)} \;=\; \|r_k^{(b)}\|. \tag{29}$$

The per-path maximum lateral deviation (normalized by chord length) is

$$\texttt{max-dev}^{(b)} \;=\; \frac{1}{\|c^{(b)}\|} \max_{0 \le k \le S} \delta_k^{(b)}. \tag{30}$$

We report the batch average:

$$\texttt{max-dev} \;=\; \frac{1}{B} \sum_{b=1}^{B} \texttt{max-dev}^{(b)}. \tag{31}$$

Smaller values indicate paths that hug the straight line more tightly (fewer detours).

**Implementation notes.** All vectors $x_k^{(b)}$ are flattened before inner products; norms are Euclidean. For numerical stability, denominators are clamped, e.g. $\|c^{(b)}\| \leftarrow \max(\|c^{(b)}\|, \varepsilon)$ and $\|s_k^{(b)}\| \leftarrow \max(\|s_k^{(b)}\|, \varepsilon)$ with $\varepsilon \approx 10^{-8}$.

## Appendix A.3 Ablation on the Blending Threshold $\tau$

To provide deeper insight into the mixed sampling strategy defined in Equation (11), we conduct an ablation study on the blending threshold $\tau$. This threshold determines the exact timestep at which the inference procedure transitions from the derived X-Flow $\tilde{v}_\theta$ to the direct V-Flow $v_\theta$.

Table 10 presents the generation performance of the SC-Flow-XL-Mix model across a range of values $\tau \in \{0.3, 0.4, 0.5, 0.6, 0.7\}$ using 250 sampling steps.

Table 10: Ablation of the blending threshold $\tau$ on ImageNet $256 \times 256$ (SC-Flow-XL-Mix).

| $\tau$ | 0.3 | 0.4 | 0.5 | 0.6 | 0.7 |
|---|---|---|---|---|---|
| **FID** | 1.89 | **1.86** | **1.86** | 1.91 | 1.93 |

The empirical results align closely with our theoretical analysis regarding sampling instability (Section 5). As the trajectory approaches the data manifold ($t \to 1$), the prediction error of the X-Flow is magnified by the factor $\frac{1}{(1-t)^2}$. Consequently, higher values of $\tau$ (e.g., 0.6 or 0.7) force the ODE solver to rely on $\tilde{v}_\theta$ in a regime where its derived velocity becomes increasingly unstable, leading to a degradation in sample quality (FID rises to 1.93).

Conversely, setting $\tau$ too low (e.g., 0.3) prematurely abandons the low-variance supervision of the endpoint predictor. By switching to the V-Flow too early, the model loses the trajectory-straightening benefits of the global destination prediction, resulting in a slight performance penalty (FID 1.89).

Optimal performance is achieved in the middle range ($\tau \in [0.4, 0.5]$). In this regime, the model maximally exploits the stable, on-manifold trajectory of the X-Flow during the highly uncertain early stages of generation, and seamlessly hands over control to the V-Flow just before the asymptotic instability of the endpoint prediction sets in. We therefore adopt $\tau = 0.5$ as a robust and theoretically justified default for our mixed inference strategy.

## Appendix A.4 Faster Sampling via Straighter Trajectories

To evaluate whether the straighter sampling trajectories induced by our consistency objective allow for faster sampling, we measure the generation quality across a reduced number of integration steps.

**Experimental Setup.** We evaluate the SC-Flow-XL model and the baseline SiT-XL model on the ImageNet $256 \times 256$ validation set. We vary the number of sampling steps $N \in \{32, 64, 128, 256\}$ using the Dopri5 ODE solver and report the resulting FID scores.

**Results and Analysis.** Figure 7 illustrate the generation quality as a function of the sampling budget. SC-Flow consistently maintains a significant performance advantage over the SiT baseline across all step counts. Notably, SC-Flow achieves an FID of 1.95 using only 128 sampling steps, which surpasses the performance of the baseline SiT-XL using 256 steps (FID 2.06).

These results empirically confirm that the improved vector field approximation and straighter ODE trajectories learned by SC-Flow directly translate to faster sampling capabilities, allowing for a substantial reduction in the computational sampling budget without sacrificing generation quality.

## Appendix A.5 2D Toy Experiment: Bridging Theory and High-Dimensional Generation

To bridge our theoretical analysis of learning errors (Section 5) with our high-dimensional image generation results, we evaluate SC-Flow on a 2D toy dataset embedded in higher ambient dimensions.

**Experimental Setup.** We generate a 2D spiral with Gaussian noise, representing a data distribution with an intrinsic dimension of $d = 2$. To simulate high-dimensional learning dynamics, we embed this 2D data into a $D$-dimensional ambient space ($D \in \{2, 8, 32\}$) using a fixed, random column-orthogonal projection

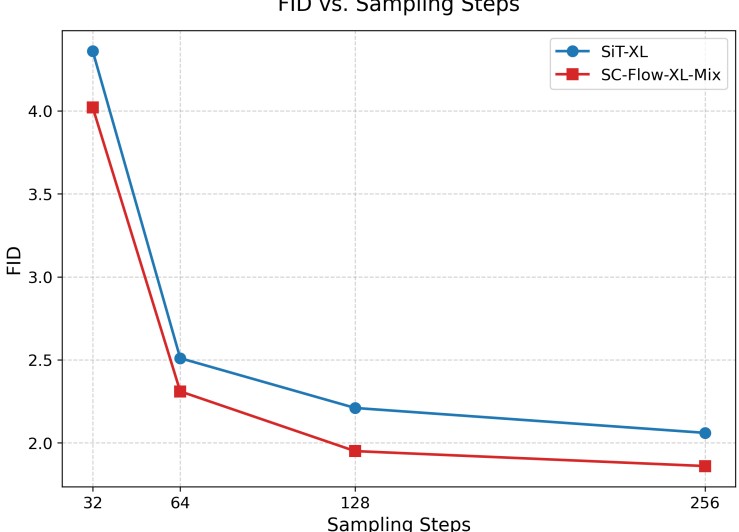

Figure 7: Generation quality (FID) versus the number of ODE sampling steps on ImageNet $256 \times 256$. SC-Flow consistently achieves better FID at lower step counts, demonstrating that its straighter trajectories enable faster sampling.

matrix. We train a lightweight generative model (a 5-layer MLP with SiLU activations and 256 hidden units) under three configurations: independent X-Flow, independent V-Flow, and our proposed SC-Flow. After training, the generated samples are projected back to the 2D subspace using the same orthogonal matrix for visualization.

**Results and Theoretical Connection.** Figure 8 illustrates the generated distributions across increasing ambient dimensions. The empirical results perfectly align with our theoretical analysis:

- **High-Variance Collapse of V-Flow:** As the ambient dimension $D$ increases, the variance of the velocity target $u = x_1 - x_0$ grows substantially, as the additional orthogonal dimensions consist entirely of pure noise. As analyzed in Section 5.1, this high-variance target space causes the independent V-Flow to struggle, eventually collapsing into unstructured noise by $D = 32$.

- **Inference Instability of X-Flow:** The independent X-Flow successfully learns the global structure across all dimensions because its target $(x_1)$ is restricted to the low-variance data manifold. However, because sampling from X-Flow requires deriving the velocity via $\tilde{v}_\theta = (\tilde{m}_\theta - x_t)/(1-t)$, it suffers from inference instability as $t \to 1$ (Section 5.2). This singularity forces the trajectories to collapse onto the mean expectation of the manifold, stripping away the natural variance (the visual "thickness") of the ground-truth data distribution.

- **Self-Consistent Flow:** SC-Flow demonstrates the regularizing power of our dual-target framework. By supervising the shared network with the stable, low-variance X-Flow target, the network is heavily regularized against high-dimensional noise. While the independent V-Flow completely collapses at $D = 32$, the V-branch of SC-Flow successfully captures the underlying spiral structure. Although the high ambient variance still introduces some noise and degradation into the SC-Flow result at $D = 32$, the consistency loss clearly rescues the velocity prediction from catastrophic failure while avoiding the asymptotic division singularity of independent X-Flow.

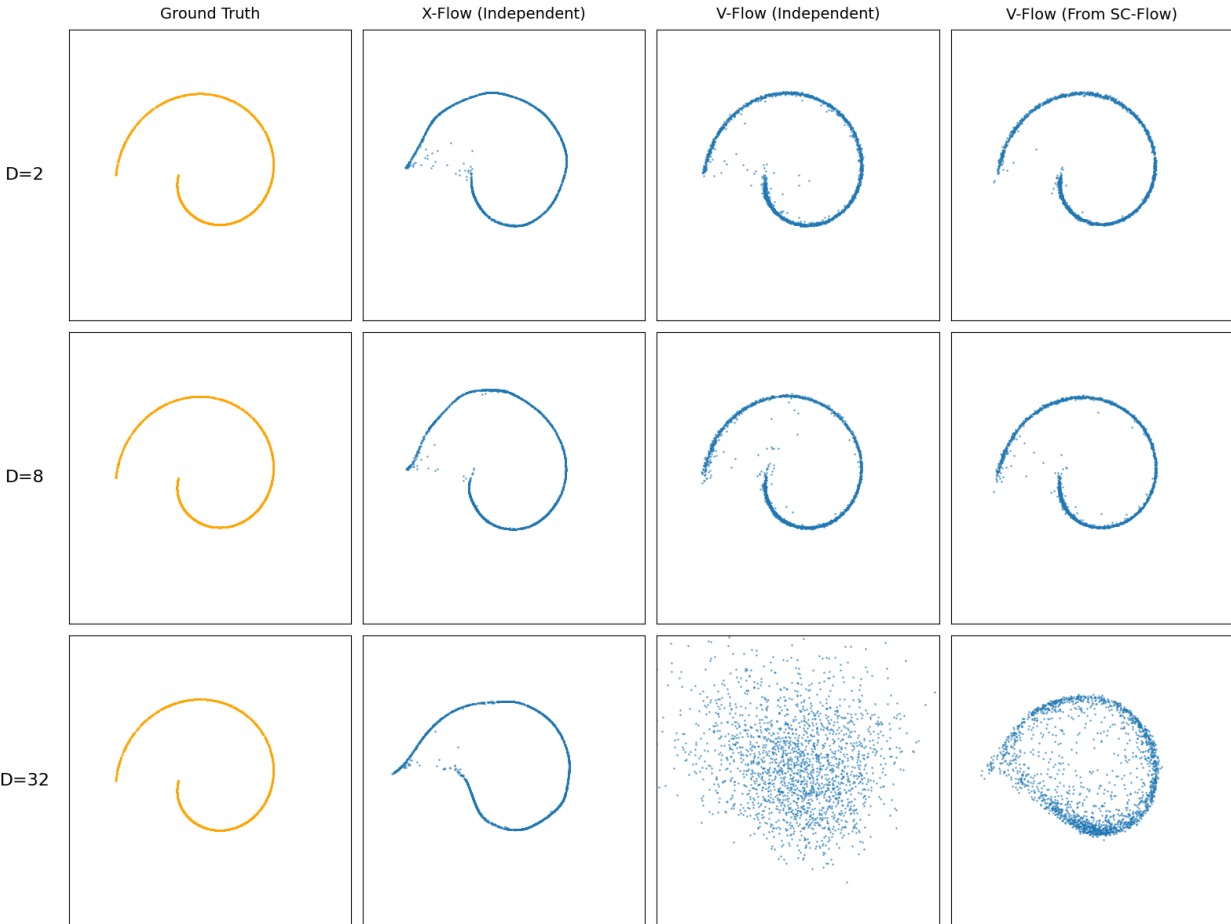

Figure 8: Toy Experiment: A $d = 2$ noisy spiral embedded in a $D$-dimensional space via an orthogonal projection matrix. As $D$ increases to 32, independent V-Flow fails due to high-variance targets. Independent X-Flow survives but loses the intrinsic data variance (thickness) due to the $t \rightarrow 1$ sampling singularity. SC-Flow (sampled via its V-output) leverages the stable X-Flow target to regularize the V-branch, successfully preventing the catastrophic collapse seen in independent V-Flow and preserving the broader manifold structure.

## Appendix A.6    Detailed Comparison with Dual-Output Diffusion Models

While SC-Flow shares the high-level philosophy of dual-target prediction with the pioneering work of Benny & Wolf (2022) on Dynamic Dual-Output Diffusion Models, the two approaches differ fundamentally in their mathematical domains, architectural implementations, and training constraints. We detail these key technical distinctions below.

**Mathematical Framework.** Benny & Wolf (2022) operate within the standard Denoising Diffusion Probabilistic Model (DDPM) (Ho et al., 2020) framework, where the network is trained to predict the added noise ($\epsilon$) and the clean data ($x_0$) to reverse a Gaussian diffusion trajectory. In contrast, SC-Flow is formulated within the continuous-time Rectified Flow (Liu et al., 2022) framework. Our model learns a deterministic ODE vector field by predicting the instantaneous velocity ($v$) and the data endpoint ($x_1$) along a straight, linear probability flow path.

**Strict Algebraic Consistency vs. Multi-Task Learning.** The most significant technical difference lies in how the dual targets are constrained. Benny & Wolf (2022) treat dual prediction primarily as a multi-task learning problem. While their network shares a feature-extraction backbone, the dual targets are learned via uncoupled loss functions and are not explicitly forced to mathematically equate to one another during training.

SC-Flow, conversely, binds the two predictions together using a strict algebraic consistency loss ($L_c$). Because Rectified Flow operates on linear paths, the exact analytical relationship between the endpoint and the velocity is strictly defined: $x_1 = x_t + (1 - t)v_t$. Our consistency loss explicitly penalizes deviations from this physical ODE constraint, guaranteeing that the network learns a singular, coherent vector field.

**Architectural Implementation.** To accommodate dual outputs, Benny & Wolf (2022) utilize a branched architecture where a shared network trunk splits into two separate, dedicated output heads. SC-Flow employs a true single-head architecture. Rather than splitting the network, we introduce a binary mode indicator bit ($b \in \{0, 1\}$) as a conditioning embedding. This forces the exact same output layer to dynamically switch its predictive behavior, maximizing parameter sharing and memory efficiency with zero architectural overhead.

**Inference and Asymptotic Stability.** Benny & Wolf (2022) rely on a learned or scheduled mixing weight during inference to dynamically blend their two predictions, essentially reconciling the differing beliefs of their two uncoupled heads.

In SC-Flow, because the consistency loss forces the X-Flow and V-Flow predictions to be mathematically equivalent, we do not need to blend them to resolve disagreements. Instead, our inference strategy is driven entirely by numerical stability. As proven in Section 5, deriving the velocity from an endpoint predictor involves an unavoidable $1/(1 - t)$ scaling factor. The transition to the direct V-Flow output at late timesteps (e.g., our switch at $\tau = 0.5$) is not a blending heuristic, but a strict numerical intervention required to bypass the asymptotic singularity of X-Flow as the trajectory approaches the data manifold ($t \to 1$).

## Appendix A.7  Pseudo-code of SC-Flow

---

**Algorithm 1** SC-Flow Training: PyTorch-like Pseudo-code

```python
class SC-FlowTrainer(nn.Module):
    def __init__(self, model, vae, transport, lr=1e-4, ema_decay=0.9999):
        super().__init__()
        self.net = model # single-head, b in {0,1}
        self.vae = vae.eval() # frozen encoder
        self.transport = transport # provides path plan, time sampling
        self.opt = torch.optim.AdamW(self.net.parameters(), lr=lr, weight_decay=0.0)
        self.ema = copy.deepcopy(self.net).eval()
        self.ema_decay = ema_decay
        torch.backends.cuda.matmul.allow_tf32 = True
        torch.backends.cudnn.allow_tf32 = True

    @torch.no_grad()
    def _update_ema(self):
        for p_ema, p in zip(self.ema.parameters(), self.net.parameters()):
            p_ema.mul_(self.ema_decay).add_(p, alpha=1.0 - self.ema_decay)

    def training_step(self, x_img, y, w_v=1.0, w_x=1.0, w_c=0.1, time_weight=False):
        with torch.no_grad():
            z = self.vae.encode(x_img).latent_dist.sample().mul_(0.18215) # (B,4,32,32)

        # sample t, x0, x1, and path (xt, ut) from transport
        t, x0, x1 = self.transport.sample(z) # shapes match z
        t, xt, ut = self.transport.path_sampler.plan(t, x0, x1)

        b0 = torch.zeros(xt.size(0), device=xt.device, dtype=torch.long) # velocity branch
        b1 = torch.ones (xt.size(0), device=xt.device, dtype=torch.long) # endpoint branch
        v_pred = self.net(xt, t, y=y, b=b0) # local motion
        x1_pred = self.net(xt, t, y=y, b=b1) # global endpoint

        L_v = ((v_pred - ut) ** 2).mean(dim=(1,2,3)).mean()
        if time_weight:
            tw = 0.5 + t.view(-1,1,1,1)
            L_x1 = (tw * (x1_pred - x1) ** 2).mean(dim=(1,2,3)).mean()
        else:
            L_x1 = ((x1_pred - x1) ** 2).mean(dim=(1,2,3)).mean()

        t_exp = t.view(-1,1,1,1)
        x1_from_v = xt + (1 - t_exp) * v_pred
        L_c = ((x1_pred - x1_from_v) ** 2).mean(dim=(1,2,3)).mean()

        loss = w_v * L_v + w_x * L_x1 + w_c * L_c
        return loss, {'L_v': L_v, 'L_x1': L_x1, 'L_c': L_c}

    def train_loop(self, loader, epochs, log_every=100):
        self.net.train()
        for ep in range(epochs):
            for i, (x_img, y) in enumerate(loader):
                x_img, y = x_img.cuda(non_blocking=True), y.cuda(non_blocking=True)
                loss, logs = self.training_step(x_img, y)
                self.opt.zero_grad(set_to_none=True)
                loss.backward()
                self.opt.step()
                self._update_ema()
                if (i + 1) % log_every == 0:
                    _ = (loss.item(), logs['L_v'].item(), logs['L_x1'].item(), logs['L_c'].item())
```

---

**Algorithm 2** SC-Flow-mix Sampling: PyTorch-like Pseudo-code

```python
class SC-FlowSampler:
    def __init__(self, model, vae, transport, cfg_scale=1.0, tau=0.5):
        self.net = model.eval()
        self.vae = vae.eval() # frozen decoder (optional for image output)
        self.transport = transport
        self.cfg_scale = cfg_scale
        self.tau = tau # switch between endpoint-induced vs direct motion

    @torch.no_grad()
    def _drift(self, x, t, y):
        if t[0] <= self.tau:
            b = torch.ones(x.size(0), device=x.device, dtype=torch.long) # endpoint branch
            x1_pred = self.net(x, t, y=y, b=b)
            t_safe = torch.clamp(1 - t, min=1e-3)
            v = (x1_pred - x) / path.expand_t_like_x(t_safe, x) # endpoint-induced motion
        else:
            b = torch.zeros(x.size(0), device=x.device, dtype=torch.long) # velocity branch
            v = self.net(x, t, y=y, b=b)
        return v

    @torch.no_grad()
    def _drift_cfg(self, x, t, y):
        if self.cfg_scale <= 1.0:
            return self._drift(x, t, y)
        half = x[: len(x)//2]
        xin = torch.cat([half, half], dim=0)
        vin = self._drift(xin, t, y)
        eps, rest = vin[:, :3], vin[:, 3:]
        cond, uncond = torch.chunk(eps, 2, dim=0)
        guided = torch.cat([uncond + self.cfg_scale * (cond - uncond)]*2, dim=0)
        return torch.cat([guided, rest], dim=1)

    @torch.no_grad()
    def sample(self, B, y, steps=250, method='euler'):
        z = torch.randn(B, 4, 32, 32, device=y.device)
        t0, t1 = self.transport.check_interval(self.transport.train_eps, self.transport.sample_eps, sde=
            False, eval=True)
        if method == 'euler':
            dt = (t1 - t0) / steps
            for k in range(steps):
                t_scalar = t0 + k * dt
                t = torch.full((B,), t_scalar, device=z.device)
                v = self._drift_cfg(z, t, y)
                z = z + v * dt
        else:
            solver = ODESolver(self._drift_cfg, t0=t0, t1=t1, steps=steps) # placeholder
            z = solver.integrate(z, y)
        x = self.vae.decode(z / 0.18215) # optional decode
        return x, z
```

## Appendix A.8 Additional Visual Results for CIFAR-10

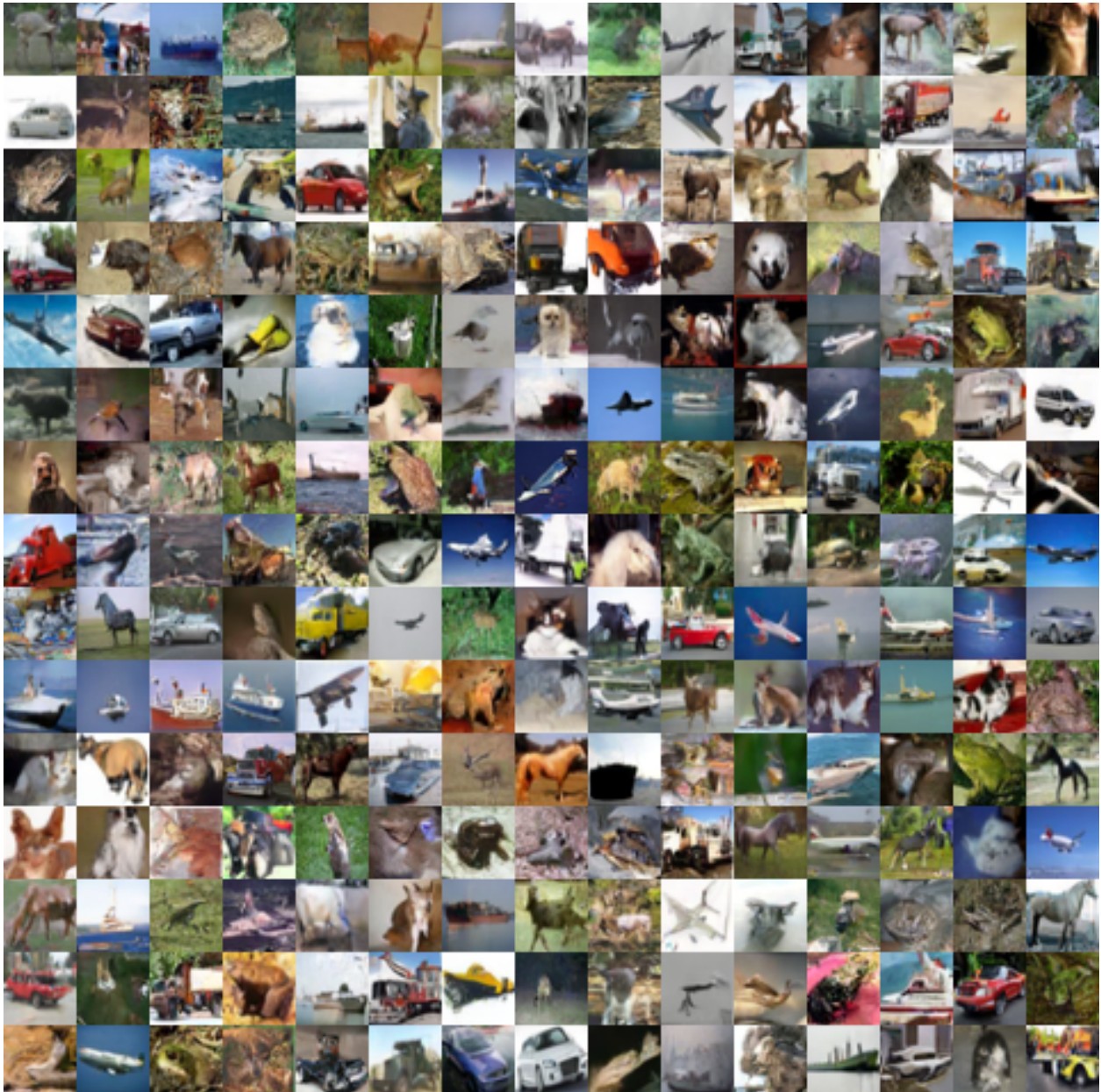

Figure 9: Samples for CIFAR-10.

## Appendix A.9 Additional Visual Results for Imagenet

All samples are from SC-Flow-mix-XL, with cfg=4.0.

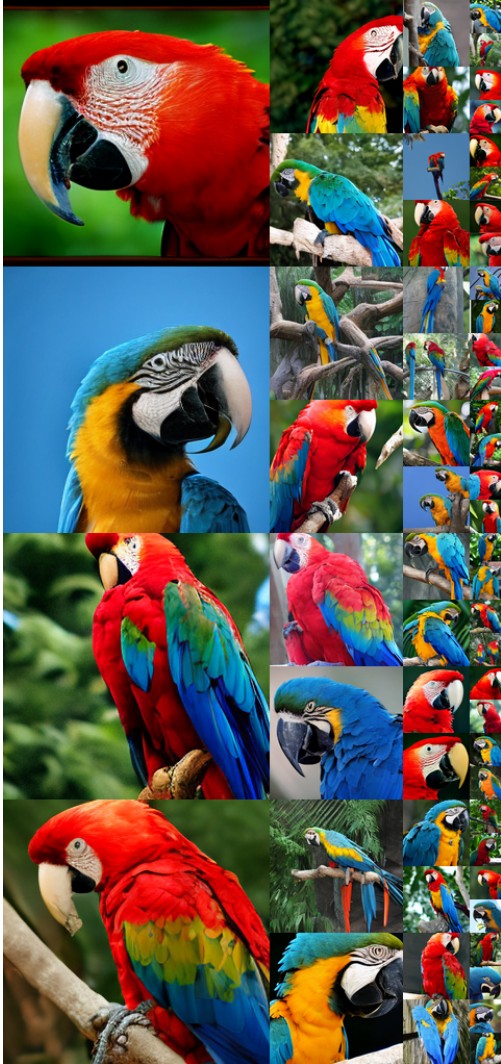

Figure 10: Samples for ImageNet class 88 (macaw).

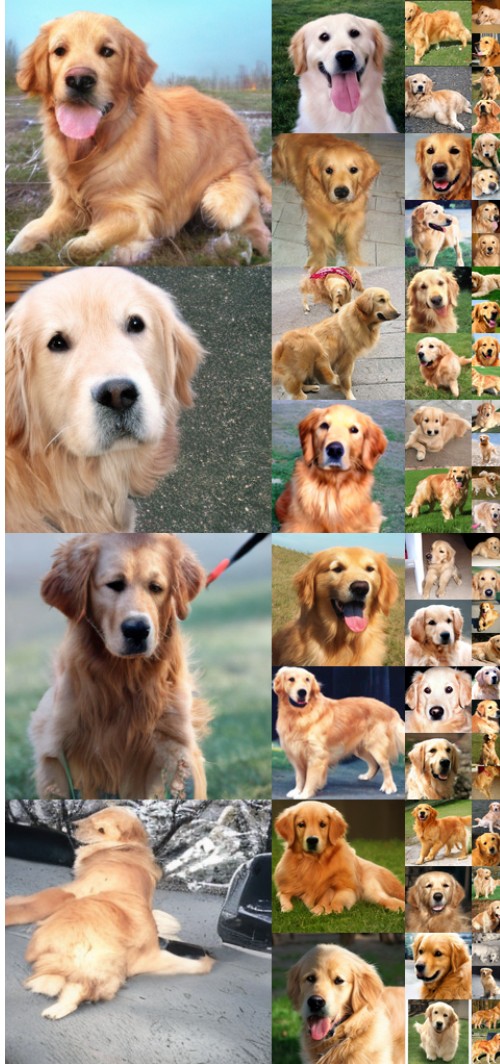

Figure 11: Samples for ImageNet class 207 (golden retriever).

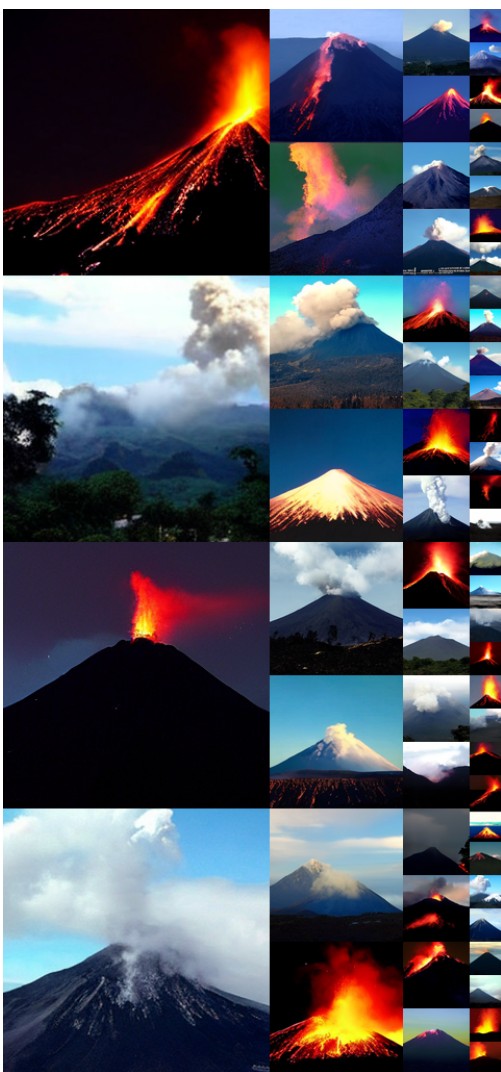

Figure 12: Samples for ImageNet class 980 (volcano).

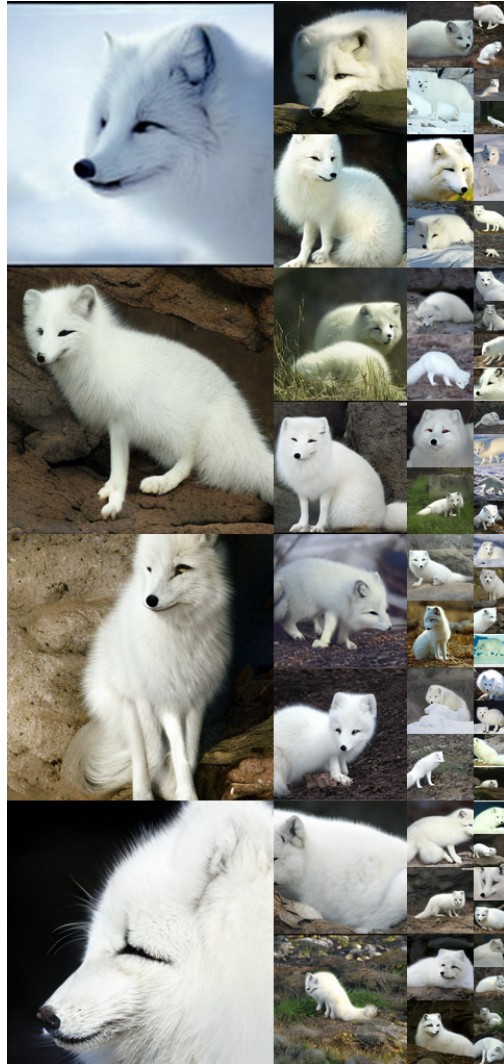

Figure 13: Samples for ImageNet class 279 (arctic fox).

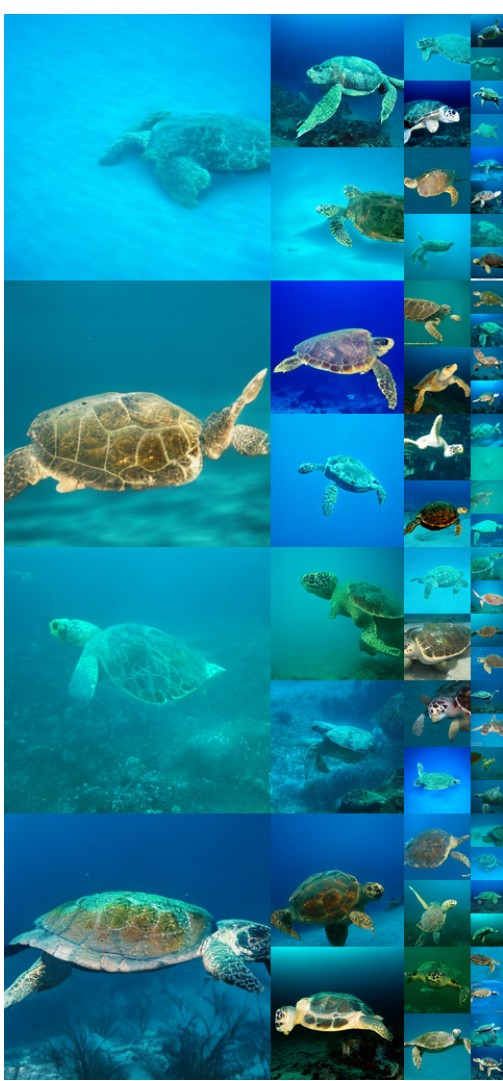

Figure 14: Samples for ImageNet class 33 (loggerhead turtle).

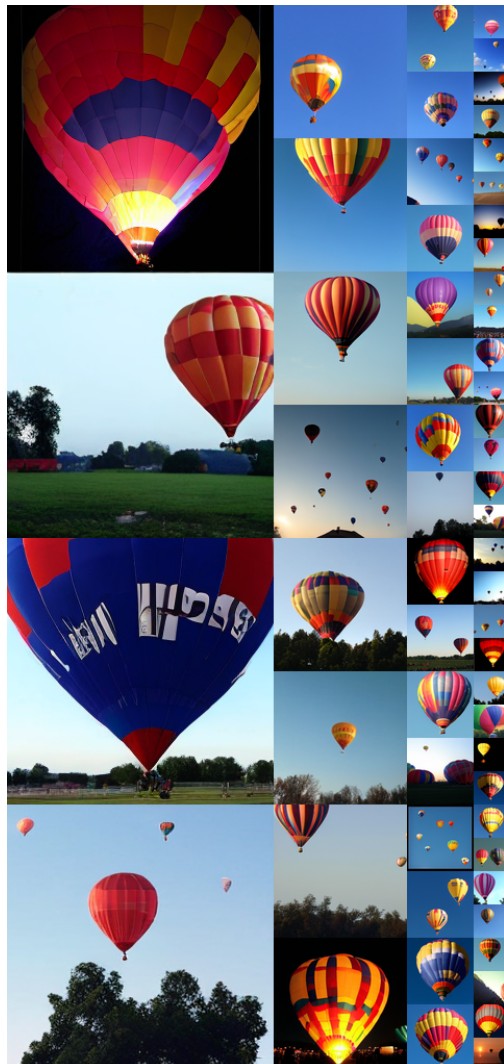

Figure 15: Samples for ImageNet class 417 (balloon).

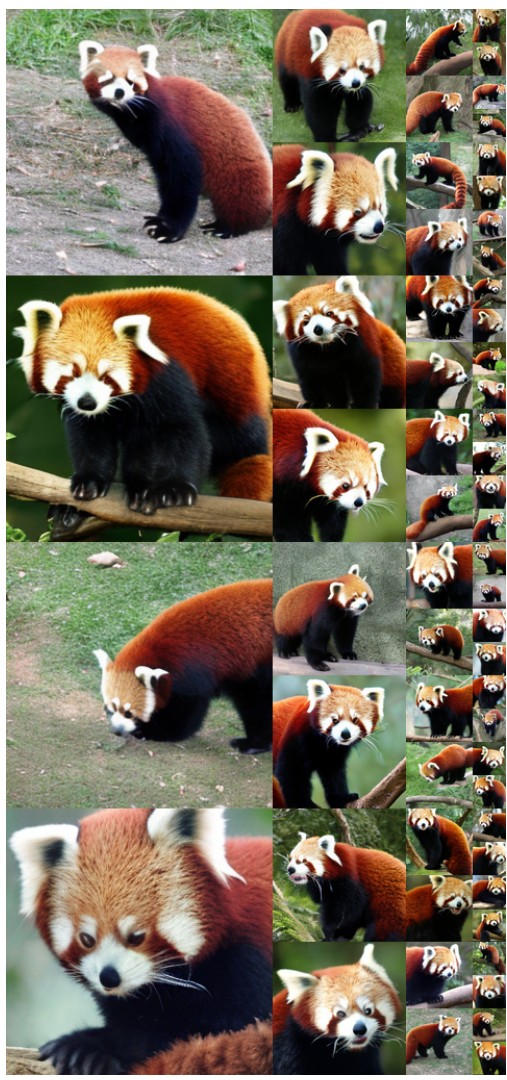

Figure 16: Samples for ImageNet class 387 (red panda).

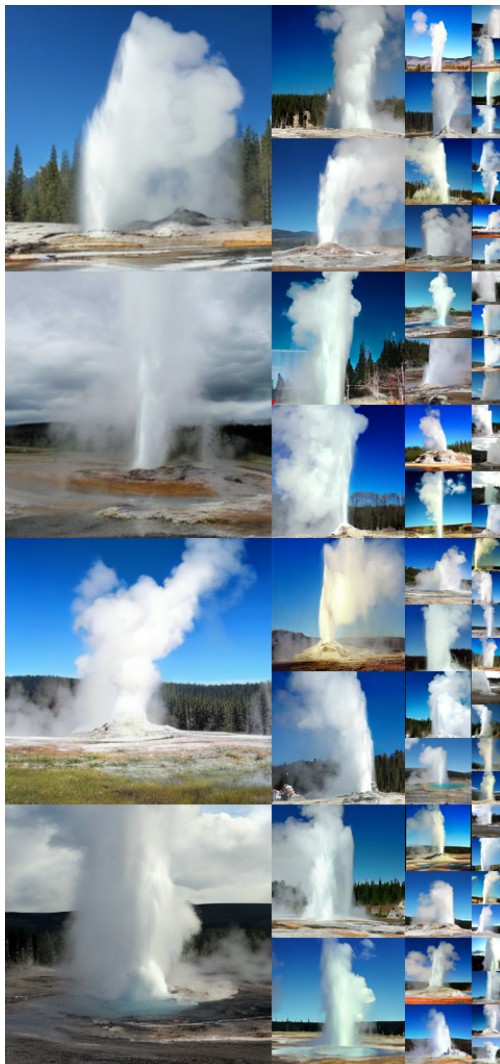

Figure 17: Samples for ImageNet class 974 (geyser).

