# OpenReview forum: "Self-Consistent Flow: Unifying Velocity and Endpoint Prediction for Rectified Flow Models"
_TMLR — Accepted by TMLR_

### Review · Reviewer_QkZg · 2026-04-11

**Summary Of Contributions:**

The paper analyzes the characteristics of different optimization objectives for flow matching, namely, predicting the velocity (v-pred) and predicting the clean data point (x-flow). The analysis reveals that x-flow stabilizes training, while v-pred leads to more stable sampling in later timesteps. The authors propose Self-Consistent Flow (SC-Flow) that combines x-flow and v-pred, and utilizes a consistency objective to align them. SC-Flow demonstrates straighter sampling trajectories, improving generation quality.

**Additional Comments:**

**Strengths**:
* The paper is overall clear and easy-to-follow (with some comments, see “Weaknesses”).
* Rigorous analysis of the different training objectives.
* I appreciate the “PyTorch-style” code blocks in the appendix.
* Sound improvement over baselines.


**Weaknesses**:
* Writing:
  * What is the difference between “x-flow” and “X-Flow”? Are they used interchangeably?
  * The analysis section starts with a vague statement: “We have two reasons to prefer X-Flow: the learning target has a low variance, and the optimal velocity stays in a low-dimensional latent space.” I found it unclear and recommend changing the structure of this section.
* Figure 2 is not entirely clear (the illustration itself).
* The paper has an interesting theoretical analysis with a simple linear example. The experimental section is focused on image generation. I believe that to bridge the two, and motivate the image generation experiments, the standard practice is to demonstrate the method on 2D toy datasets, which allow for a comparison of the generation results.
* Limitations: very short discussion in the appendix, I recommend extending it. I also recommend moving some parts to the main text.


**Minor**:
* References to numbered equations, figures, appendix etc. should be “Equation 4” (e.g., page 4, just after Eq. 6, page 6 just before Eq. 13m page 7, just before Sec 6.1).
* Page 5: “Here we assume $y$ is a condition with a simple form.” - what is “a simple form?
* Figure 1 is never referenced, and includes an important illustration of the method.
* Figure 2 is never referenced. Also, missing a space between “(b)” and “Sampling”.
* The main text referenced “Theorem 3” (page 11). There is no such theorem in the paper.
* Figure 6: the title has “BiFlow” in it. I assume that is a mistake.

**Audience:**

Yes

**Audience Explanation:**

Yes, the generative modeling community would be interested in the findings of this paper.

**Broader Impact Concerns:**

This is a generative modeling paper, and as with every generative model, there are obvious risks; however, the scope of this paper is limited to well-known image datasets. I don't have a strong opinion about a "Broader Impact" statement.

**Claims And Evidence:**

Yes

**Claims Explanation:**

Yes, the claims are supported by clear evidence. See below for strengths.

**Requested Changes:**

* Question: To my understanding, straighter sampling trajectories should allow for *faster* sampling, by reducing the sampling steps. Has this been observed with SC-Flow?
* Question: Has $\tau$ in Eq. 11 been ablated? While $\tau=0.5$ seems like a reasonable value, given the analysis of the regimes, I believe more details are required.
* See “Weaknesses” and “Minor”.

---

> ### Author Response · Authors · 2026-04-18
> **Rebuttal by Authors (1)**
>
> Thank you for the detailed and constructive feedback on our paper. We appreciate that our work is recognized as having a "rigorous analysis of the different training objectives" and that it provides a "sound improvement over baselines." We are also glad that the "PyTorch-style" code blocks were helpful and clear. We take your comments seriously and have revised the manuscript accordingly. We provide the following responses, which hopefully address the issues raised.
>
> > To my understanding, straighter sampling trajectories should allow for faster sampling, by reducing the sampling steps. Has this been observed with SC-Flow?
>
> We thank the reviewer for this insightful question. Your intuition is absolutely correct. The improved vector field approximation and straighter ODE trajectories learned by SC-Flow do indeed directly translate to faster sampling capabilities.
>
> To explicitly verify this, we have conducted a new evaluation and added the results to the revised manuscript (Appendix A.4, "Faster Sampling via Straighter Trajectories"). We evaluated the SC-Flow-XL model against the SiT-XL baseline on ImageNet 256x256, measuring generation quality (FID) across a reduced number of integration steps (N = 32, 64, 128, and 256).
>
> We observed that SC-Flow consistently maintains a significant performance advantage over the baseline across all step counts. Notably, SC-Flow achieves an FID of 1.95 using only 128 sampling steps. This surpasses the performance of the baseline SiT-XL using 256 steps (FID 2.06).
>
> This result empirically confirms that SC-Flow allows for a 2x reduction in the computational sampling budget while still delivering superior generation quality compared to the standard rectified flow baseline.
>
> > Has $\tau$ in Eq. 11 been ablated? While $\tau$ seems like a reasonable value, given the analysis of the regimes, I believe more details are required.
>
> We appreciate the reviewers' suggestions and believe such ablation studies are helpful. We have added a comprehensive ablation study on the blending threshold $\tau$ to the revised manuscript (Appendix A.3). The empirical results align with our theoretical analysis: setting $\tau$ too high (e.g., 0.6 or 0.7) forces the ODE solver to rely on X-Flow in the highly unstable (t,1) regime, degrading performance. Conversely, setting $\tau$ too low (e.g., 0.3) prematurely abandons the low-variance supervision of the endpoint predictor. $\tau$=0.5 provides the optimal balance between early-stage stability and late-stage robustness.
>
> > What is the difference between “x-flow” and “X-Flow”? Are they used interchangeably?
>
> They were used interchangeably by mistake. We have standardized the terminology to "X-Flow" throughout the revised manuscripyt to avoid any confusion.
>
> > The analysis section starts with a vague statement: “We have two reasons to prefer X-Flow: the learning target has a low variance, and the optimal velocity stays in a low-dimensional latent space.” I found it unclear and recommend changing the structure of this section.
>
> Thanks for the reviewers' insightful question. We agree and have restructured Section 5 in the revision. We rewrote the introductory paragraph to clearly outline the trade-off being analyzed (training stability vs. inference instability). Furthermore, we added clear subsection titles ("5.1 Training Stability and Target Variance" and "5.2 Inference Instability Near the Data Manifold") to better guide the reader through the mathematical arguments.
>
> > Figure 2 is not entirely clear (the illustration itself).
>
> We thank the reviewer for pointing this out. Upon reflecting on this feedback, we agreed that the conceptual illustration did not add sufficient clarity to the theoretical discussion. Because the mathematical derivations and accompanying text in Section 5 are rigorous and clear enough to stand on their own, we have opted to remove Figure 2 entirely from the revised manuscript to prevent any potential confusion.

---

> > ### Author Response · Authors · 2026-04-18
> > **Rebuttal by Authors (2)**
> >
> > > The paper has an interesting theoretical analysis with a simple linear example. The experimental section is focused on image generation. I believe that to bridge the two, and motivate the image generation experiments, the standard practice is to demonstrate the method on 2D toy datasets, which allow for a comparison of the generation results.
> >
> > We thank the reviewer for this excellent suggestion. We strongly agree that a 2D toy experiment provides the perfect bridge between our mathematical analysis in Section 5 and our high-dimensional ImageNet evaluations.
> >
> > Following your recommendation, we have conducted this exact experiment and added a comprehensive new section to the revised manuscript (Appendix A.5, "2D Toy Experiment: Bridging Theory and High-Dimensional Generation").
> >
> > In this new experiment, we embedded a 2D noisy spiral into increasingly higher ambient dimensions (D = 2, 8, and 32) using an orthogonal projection matrix to evaluate independent X-Flow, independent V-Flow, and SC-Flow. The empirical results perfectly mirror our theoretical claims. Specifically, we observe that as the ambient dimension increases to D=32, independent V-Flow suffers from the high-variance noise target and completely collapses, which validates our analysis in Section 5.1. Conversely, independent X-Flow survives the noise but loses the intrinsic data variance (the "thickness" of the spiral) due to the asymptotic sampling singularity, validating Section 5.2. SC-Flow resolves this by leveraging the stable X-Flow target to regularize the V-branch. By sampling from this V-output, SC-Flow successfully survives the high dimensions while preserving the full manifold variance.
> >
> > This new visual demonstration empirically validates our theoretical trade-offs and provides a strong, intuitive motivation for our dual-target architecture before diving into the pixel-space generative results. We are very grateful for this feedback, as it has significantly strengthened the narrative of the paper.
> >
> > > Limitations: very short discussion in the appendix, I recommend extending it. I also recommend moving some parts to the main text.
> >
> > Thank you for this suggestion. We have significantly expanded the Limitations section and moved it into main text. It covers our reliance on linear paths, scaling to other modalities, training overhead, and the heuristic blending threshold.
> >
> > > References to numbered equations, figures, appendix etc. should be “Equation 4” (e.g., page 4, just after Eq. 6, page 6 just before Eq. 13m page 7, just before Sec 6.1).
> >
> > Thank you so much for carefully reading. We have corrected all references to equations, figures, and sections to follow the capitalized and spelled-out format (e.g., "Equation 4") throughout the manuscript.
> >
> > > Page 5: “Here we assume y is a condition with a simple form.” - what is “a simple form?
> >
> > We have clarified this in the revised text. By "simple form," we mean y is a global, discrete conditioning variable (such as a class label) that can be mapped to a single embedding vector and added directly to the network's intermediate states, avoiding the need for complex spatial or sequential conditioning mechanisms.
> >
> > > Figure 1 is never referenced, and includes an important illustration of the method.
> >
> > We have added references to Figure 1 throughout Section 4 of the revised manuscript to better visually anchor the mathematical and architectural descriptions of the method.
> >
> > > The main text referenced “Theorem 3” (page 11). There is no such theorem in the paper.
> >
> > This was a typo from a previous draft. We have corrected the text to accurately reference the mathematical phenomena established in "Section 5".
> >
> > > Figure 6: the title has “BiFlow” in it. I assume that is a mistake.
> >
> > Thanks for your careful reading. We have updated the figure to correctly use "SC-Flow".
> >
> > We've made updates based on the feedback provided (all revisions are marked in blue), and we believe that these changes substantially improve the manuscript. We're grateful for the valuable comments and thank you for your time and expertise.

---

> > > ### Comment · Reviewer_QkZg · 2026-06-09
> > > **Thank you for the revision**
> > >
> > > I thank the authors for their effort, I believe the changes significantly improved the papers. All my concerns have been resolved and I'm happy to recommend acceptance.

---

### Review · Reviewer_LEL5 · 2026-04-20

**Summary Of Contributions:**

This paper studies why different rectified-flow parameterizations behave differently in practice. It argues that endpoint prediction provides a lower-variance target, while direct velocity prediction is more stable near the data manifold at sampling time. Based on this trade-off, it proposes SC-Flow, a dual-target training scheme that uses one shared backbone with a mode bit and a consistency loss to couple velocity and endpoint prediction. Experiments on CIFAR-10 and ImageNet-256 show improvements relative to standard rectified-flow baselines.

Strengths:
- The method is simple and plug-and-play: one backbone and no extra sampling cost

- The empirical gains on ImageNet-256 are meaningful

- The paper offers a coherent intuition for why the method may work, tying together target variance, manifold structure, and late-time sampling stability

Weakness:
- The theoretical story relies on fairly restrictive assumptions (independent coupling, linear rectified paths, linear manifold view), so it is more a stylized explanation than a general account of rectified-flow behavior.

- The empirical scope is still relatively narrow given the broad framing: mostly class-conditional image generation on CIFAR-10 and ImageNet-256

**Audience:**

Yes

**Audience Explanation:**

The choice of target parameterization in flow/ODE-based generative models is an active and practically important topic, and this paper offers both a simple method and a useful explanatory angle.

**Claims And Evidence:**

Yes

**Claims Explanation:**

The central empirical claims are mostly supported. The theoretical analysis is derived under simplified assumptions, and the experiments do not yet show that the same explanation carries over broadly beyond the specific rectified-flow image-generation regime studied here.

**Requested Changes:**

- Some typo/mismatches = between Figure 2 (“ImageNet-21k”) and other places (ImageNet-1K), and the inconsistency around the L/4 training budget (400K vs. 500K)

- Tighten the scope of the theoretical claims. Please make explicit that the variance/manifold arguments are derived under simplified assumptions and should not be presented as a full explanation of real-world rectified-flow behavior without qualification

---

> ### Author Response · Authors · 2026-05-02
>
> We sincerely thank the reviewer for the thoughtful and positive review. We appreciate the comment that SC-Flow is a simple, plug-and-play method and that our theoretical intuition regarding target variance and sampling stability resonated with you. We also appreciate your feedback on our theoretical scope.
>
> >Some typo/mismatches = between Figure 2 (“ImageNet-21k”) and other places (ImageNet-1K), and the inconsistency around the L/4 training budget (400K vs. 500K)
>
> Thank you for catching these typos. We have corrected the caption of Figure 2 and the dataset description in the appendix to accurately reflect that the model was trained on the ImageNet 256x256 training set. Additionally, we have resolved the inconsistency regarding the training budget for the DiT-L/4 model. The correct training budget is 400K iterations (as correctly shown in Table 2), and we have corrected the number in Section 6.2.
>
> >Tighten the scope of the theoretical claims. Please make explicit that the variance/manifold arguments are derived under simplified assumptions and should not be presented as a full explanation of real-world rectified-flow behavior without qualification.
>
> Thank you for this constructive feedback. In Section 5, based on the suggestion, we have made it more clear for our general assumption of the setting with rectified flow: independent coupling, straight conditional paths, and linear noise scheduling. While it is one setting of a large space of possible settings, we also see some practical methods that directly use this setting (e.g. [1]). For the analysis with linear projection, we also added the assumption in the revised text: the data can be encoded in a low-dimensional linear space. We agree that the linear assumption is rarely true in real data, but it still serves as an instrumental tool for the analysis of
> various learning models. We have explicitly added a clarification in the text.
>
> We have also expanded the Limitations section to formally discuss the restrictions of our linear path assumption, noting that extending these explanations to non-linear flow matching trajectories or alternative manifolds remains an open challenge.
>
> We have updated our paper to incorporate your comments. We appreciate your feedback and are happy to address further concerns.
>
> [1] Esser, Patrick, et al. "Scaling Rectified Flow Transformers for High-Resolution Image Synthesis." International Conference on Machine Learning. PMLR, 2024.

---

### Review · Reviewer_8UVV · 2026-05-25

**Summary Of Contributions:**

The authors argue that in the formulation of rectified flow models, endpoint prediction provides a lower-variance and more stable training signal, while velocity prediction is more stable during sampling especially near the data manifold. Based on this view, the paper proposes self-consistent flow (SC-Flow), which trains a single model to predict both the local velocity and the endpoint. Experimental results show improved generation quality over rectified flow baselines, including better FID scores and faster training convergence.

**Additional Comments:**

Minor points:
- Line 7 of Algorithm 2: What is the $\operatorname{mix}$ operator? Is it just the rule in Eq. (11)?
- Table 1, IS for SC-Flow-Mix ($\tau=$0.5): typo of 10.65?
- Figure 5, caption: What is Theorem 3 mentioned there?

**Audience:**

Yes

**Audience Explanation:**

Flow matching / diffusion models are of great interest in the community, and the paper presents an interesting view on these subjects.

**Claims And Evidence:**

Yes

**Claims Explanation:**

Overall, I find the paper well motivated and technically coherent. The proposed method is based on a clear observation about the complementary strengths of velocity and endpoint prediction, and the theoretical discussion, training objective, and empirical evaluation are generally aligned with this motivation. The experimental results support the main claims, and the ablation studies help clarify the role of the consistency loss and the shared architecture.

**Requested Changes:**

My main concern is that the relationship to Benny & Wolf (2022), Dynamic Dual-Output Diffusion Models, should be discussed in more detail. Although the base idea to combine endpoint and velocity predictions is common, the current description in the Related Work section remains minimal. The key technical differences and resulting (anticipated) behaviors should be elaborated more, even deferring some details to a section in the appendix. I think this is important for placing the contribution more correctly in the context of related studies.

---

> ### Author Response · Authors · 2026-05-28
> **Rebuttal by Authors**
>
> We sincerely thank the reviewer for the thoughtful and positive review. We are very glad you found the paper well-motivated, technically coherent, and that our theoretical discussion aligns well with the proposed method and empirical results. We appreciate your constructive feedback regarding related work and the minor typos, which have helped us improve the clarity and precision of the manuscript.
>
> > My main concern is that the relationship to Benny & Wolf (2022), Dynamic Dual-Output Diffusion Models, should be discussed in more detail.
>
> We agree that more discussion of the relationship between our work and Benny & Wolf (2022) will better place our work in the literature.
>
> In section 4, we have added a paragraph "As discussed in Section 2 ..." to compare our method against Benny & Wolf immediately after we have introduced our method. We have also added a dedicated section in Appendix A.6 ("Detailed Comparison with Dual-Output Diffusion Models") to elaborate on these distinctions.
>
> Specifically, we highlight that rather than treating dual prediction as a multi-task problem with independent heads, SC-Flow utilizes a single-head mode-bit architecture and binds the outputs with a strict algebraic consistency loss based on the physical ODE constraint ($x_1 = x_t + (1-t)v_t$). Furthermore, we clarify that our piecewise sampling strategy is not a heuristic to blend conflicting outputs, but a required numerical intervention to bypass the asymptotic singularity of X-Flow as $t \to 1$.
>
> > Line 7 of Algorithm 2: What is the operator? Is it just the rule in Eq. (11)?
>
> Yes, your understanding is correct. It represents the mixing rule defined in Equation 11. We have updated Line 7 in Algorithm 2 to explicitly read: "$x_t \leftarrow \text{mix}(v_\theta, \tilde{v}_\theta, t)$ as described in Equation (11)".
>
> > Table 1, IS for SC-Flow-Mix (0.5): typo of 10.65?
>
> Thank you for catching this. Yes, 0.65 was indeed a transcription typo. We have corrected the Inception Score to 10.65 in the revised Table 1.
>
> > Figure 5, caption: What is Theorem 3 mentioned there?
>
> We apologize for the confusion; "Theorem 3" was a typo left over from an earlier structural draft of the paper. We have corrected the caption of Figure 5 to accurately reference the "inference stability analysis in Section 5".
>
> We have updated our paper to incorporate your comments. We appreciate your feedback and are happy to address further concerns.

---

### Decision · Action_Editor_ZMTV · 2026-07-01

**Recommendation:** Accept as is

**Audience:**

Yes

**Audience Explanation:**

Flow matching and diffusion-based generative models are active areas of research and of broad interest to the machine learning community. The paper provides useful theoretical and empirical insights into these methods, and the proposed framework is likely to be of interest to researchers working on generative modeling and related topics.

**Claims And Evidence:**

Yes

**Claims Explanation:**

This paper studies different objectives for flow matching and proposes SC-Flow, a framework that combines endpoint prediction and velocity prediction within a unified model. The paper is  well motivated and technically sound, with theoretical analysis and experimental results that support the main claims. The empirical evaluation shows improvements over relevant baselines, and the ablation studies help clarify the contribution of different components. During revision, the authors addressed the concerns satisfactorily.